# TENT5-mediated polyadenylation of mRNAs encoding secreted proteins is essential for gametogenesis in mice

Michał Brouze[1,2], Agnieszka Czarnocka-Cieciura[1], Olga Gewartowska[1,3,4], Monika Kusio-Kobiałka[1], Kamil Jachacy [1,4], Marcin Szpila [3,5], Bartosz Tarkowski[1,2], Jakub Gruchota[1,2], Paweł Krawczyk [1,2], Seweryn Mroczek[1,4], Ewa Borsuk[1,5] & Andrzej Dziembowski [1,2,4,5] ✉

Cytoplasmic polyadenylation plays a vital role in gametogenesis; however, the participating enzymes and substrates in mammals remain unclear. Using knockout and knock-in mouse models, we describe the essential role of four TENT5 poly(A) polymerases in mouse fertility and gametogenesis. TENT5B and TENT5C play crucial yet redundant roles in oogenesis, with the double knockout of both genes leading to oocyte degeneration. Additionally, TENT5B-GFP knock-in females display a gain-of-function infertility effect, with multiple chromosomal aberrations in ovulated oocytes. TENT5C and TENT5D both regulate different stages of spermatogenesis, as shown by the sterility in males following the knockout of either gene. Finally, *Tent5a* knockout substantially lowers fertility, although the underlying mechanism is not directly related to gametogenesis. Through direct RNA sequencing, we discovered that TENT5s polyadenylate mRNAs encoding endoplasmic reticulum-targeted proteins essential for gametogenesis. Sequence motif analysis and reporter mRNA assays reveal that the presence of an endoplasmic reticulum-leader sequence represents the primary determinant of TENT5-mediated regulation.

Essentially, every eukaryotic mRNA possesses a poly(A) tail. Bound by poly(A) binding proteins (PABPs), the poly(A) tail facilitates mRNA export from the nucleus and enhances protein synthesis through interactions with translation initiation factors. The poly(A) tail also stabilizes mRNA molecules by preventing exoribonucleolytic decay. Consequently, their deadenylation rate largely determines mRNA half-life. When a poly(A) tail in the cytoplasm is shortened to ~20 nucleotides (nt), PABP is released, rendering mRNA translationally inactive and susceptible to degradation. However, in specific contexts, deadenylated mRNA is stored in a dormant state to be later readenylated in the cytoplasm, to activate protein synthesis[1–4]. Such non-canonical cytoplasmic polyadenylation has been mostly studied in the context of

gametogenesis[1–4] and local translation at synapses[5,6]. In these instances, certain mRNAs are rapidly polyadenylated in response to cellular signals, allowing translation to start in a transcription-independent fashion. Our knowledge of cytoplasmic polyadenylation comes primarily from either biochemical analysis in *Xenopus laevis* oocytes or genetic studies in invertebrates. The first and only extensively studied cytoplasmic poly(A) polymerase is GLD2 (or TENT2 according to new nomenclature)[3,5,7–14]. In non-mammalian species, including *Caenorhabditis elegans*, *Xenopus laevis*, and *Drosophila melanogaster*, TENT2 clearly participates in the elongation of poly(A) tails in gametes and early embryos[3,4,10,15]. TENT2 is also important for synaptic plasticity in the fruit fly[5]. On its own, TENT2 does not contain a detectable RNA-

[1]Laboratory of RNA Biology, International Institute of Molecular and Cell Biology, Warsaw 02-109, Poland. [2]Institute of Biochemistry and Biophysics, Polish Academy of Sciences, Warsaw 02-106, Poland. [3]Genome Engineering Facility, International Institute of Molecular and Cell Biology, Warsaw 02-109, Poland. [4]Institute of Genetics and Biotechnology, Faculty of Biology, University of Warsaw, Warsaw 02-106, Poland. [5]Laboratory of Embryology, Institute of Developmental Biology and Biomedical Research, Faculty of Biology, University of Warsaw, Warsaw 02-096, Poland. ✉e-mail: adziembowski@iimcb.gov.pl

binding domain and in order to select substrates, it cooperates with several RNA-binding proteins such as GLD3 in *C. elegans*[4]. Seminal work on *X. laevis* oocytes has led to a model of cytoplasmic polyadenylation. In this model, cycles of deadenylation and TENT2-mediated polyadenylation are regulated by the cytoplasmic polyadenylation element binding protein 1 (CPEB1), in cooperation with several other factors[1]. Data has suggested that TENT2 is also involved in cytoplasmic polyadenylation in mammals. This is further supported by experiments in which human TENT2, tethered to a reporter mRNA and injected into *X. laevis* oocytes, activated translation. In line with this hypothesis, the knock-down or overexpression of TENT2 in mouse oocytes results in delayed maturation and frequent arrest in metaphase I[16]. However, TENT2 knockout (KO) mice of both sexes are fertile and display no major phenotype[17]. Additionally, oocyte maturation is normal, and the poly(A) tail length of reporter mRNA is not altered in germline or somatic cells[17]. Moreover, TENT2 KO mice do not exhibit any behavioral abnormalities, suggesting that polyadenylation in neurons is also unaffected[7]. This raises the possibility that in mammals, other TENT protein(s), yet to be identified, could be involved in poly(A) tail length regulation. However, all other previously described mammalian non-canonical poly(A) polymerases are either mitochondrial (mtPAP)[18] or mostly nuclear (TENT4A and TENT4B)[18–21], making them unlikely to play a significant role in cytoplasmic polyadenylation. In contrast to TENT2, its potential mammalian regulators, CPEBs (1–4), are more comprehensively studied[22]. Indeed, CPEB1 KO mice display severe gametogenesis defects[23] as well as impaired long-term potentiation in neurons[24]. The three other CPEB proteins contribute to the regulation of various physiological processes, although in many cases, they function as translational repressors rather than activators[25]. Thus, the mechanisms, and impact of cytoplasmic polyadenylation during gametogenesis remain to be established.

A few years ago, we described a previously overlooked family of non-canonical poly(A) polymerases: TENT5 (FAM46)[26]—all members of which were predicted to contain a putative nucleotidyltransferase catalytic domain[26,27]. In mammals, this family of cytoplasmic proteins has four members (TENT5A, TENT5B, TENT5C, and TENT5D)[27] that are differentially expressed in tissues and organs. TENT5A is involved in osteogenesis[28], TENT5C enhances expression of immunoglobulin mRNA in B cells[29], while TENT5A and TENT5C together regulate the expression of innate immune response proteins in macrophages[30]. Here, we analyzed gametogenesis-related phenotypes of all four *Tent5* KO mutations. We demonstrate the involvement of TENT5B and TENT5C in oogenesis. We further show that TENT5C and TENT5D participate in spermatogenesis, which was also recently identified by others[31–34]. Using direct RNA sequencing, we identified TENT5 substrates in both ovaries and testes. The most prominent mRNAs regulated by TENT5 encode secreted proteins, many of which are essential for gametogenesis such as zona pellucida components produced by oocytes. The leader sequence for the endoplasmic reticulum (ER) is also sufficient for TENT5-mediated poly(A) tail length regulation in oocytes, pointing to a simple mechanism of substrate recognition.

## Results

### TENT5 poly(A) polymerases are involved in both spermatogenesis and oogenesis

To understand the physiological role of the TENT5 family of cytoplasmic poly(A) polymerases, we generated and analyzed genetically modified mouse lines carrying constitutive knockout (KO) mutations in all members of the *Tent5* family. The *Tent5a* and *Tent5c* KO mouse lines were described previously[26,28]. We simultaneously developed two *Tent5b* KO mouse lines—one that has an 11 bp deletion, leading to a p.S119EfsX15 frameshift mutation, and another with a deletion spanning the complete catalytic center of the protein (p.L121P, G122_S238del). Both mouse lines exhibited an identical phenotype;

hence, they were used interchangeably in this study. The *Tent5d* KO mouse line has a del36bp, ins6bp mutation resulting in the deletion of ten amino acids, including D82 and D84, which are critical for TENT5D activity, and the insertion of two amino acids (p.76-85del, insCL). Furthermore, due to the lack of reliable, commercially available specific antibodies against TENT5 proteins, we generated several new knock-in mouse lines with either endogenous TENT5B or TENT5D tagged with green fluorescent protein (GFP) or FLAG tags. We also used previously generated FLAG- and GFP-tagged TENT5C mouse lines[26,29].

Individual KOs of all *Tent5* genes did not impact the development and birth ratio in matings of heterozygous males and females in each mouse line. In the case of *Tent5b/c* double knockout (dKO), both parents in the tested mating had *Tent5b*$^{-/-}$ *Tent5c*$^{+/-}$ genotype. Since *Tent5d* is located on the X chromosome, homozygous KO females could not be generated due to male infertility described further (Fig. 1A, *Tent5a* data previously reported by Gewartowska, et al.[28]). Additionally, these mutations did not affect overall weight (except for *Tent5a* KO previously reported by Gewartowska, et al.[28]; Supplementary Fig. 1A) or blood morphology (except for *Tent5c* KO previously reported by Mroczek et al.[26]; Supplementary Fig. 1B).

During the initial breeding process, while establishing KO mouse lines, we observed that homozygotic KO mutations of several *Tent5* genes caused partial or complete infertility in both males and females (Fig. 1B, Table 1). This comprised complete male, but not female, infertility in *Tent5c* KO (*Tent5c*$^{-/-}$), male infertility in *Tent5d* KO (*Tent5d*$^{-/null}$), partial male infertility and complete infertility in *Tent5a* KO (*Tent5a*$^{-/-}$) animals, and complete infertility in *Tent5b/c* double KO (dKO; *Tent5b*$^{-/-}$ *Tent5c*$^{-/-}$) females (with one wild-type (WT) allele of either gene sufficient for normal fertility). The presence of the GFP tag at the C-terminus of TENT5B (TENT5B-GFP) also impacted fertility—homozygous *Tent5b*$^{gfp/gfp}$ females were completely infertile, while *Tent5b*$^{wt/gfp}$ females had reduced fertility. In contrast, no such effect was observed for N-terminally tagged TENT5B (GFP-TENT5B).

Despite the partial infertility associated with *Tent5a* KO, the morphology of male sperm and testis was normal. Additionally, we observed several cases of fertile *Tent5a* KO males. In females, ovarian morphology (Supplementary Fig. 1C) and histological structure (Supplementary Fig. 1D) remained normal, and they produced a healthy pool of oocytes at the germinal vesicle (GV) stage (Supplementary Fig. 1E). Furthermore, isolated GV oocytes could mature into metaphase II oocytes in vitro (Fig. 1C). Taken together, these results show that the subfertility of *Tent5a* KO animals is not directly related to the process of gametogenesis itself, but rather to the poor general condition of animals, as previously reported[28].

We conclude that TENT5C and TENT5D are essential for spermatogenesis, whereas TENT5B and TENT5C play a crucial but redundant role in oogenesis.

### Lack of TENT5B and TENT5C leads to early oogenesis arrest

As mentioned previously, a complete double KO of *Tent5b* and *Tent5c* results in female infertility, contrasting the lack of phenotypes in mice with disrupted *Tent2* (*Gld2*)[17]. Even one WT allele of either *Tent5b* or *Tent5c* is sufficient to maintain normal fertility, pointing to their redundant roles. Accordingly, both genes are highly expressed in GV oocytes, as observed in the oocytes of *Tent5c*$^{GFP}$ and *Tent5b*$^{GFP}$ knock-in females expressing respective C-terminal GFP fusions (Fig. 2A). When we examined 8-week-old *Tent5b/c* dKO female mice, we detected markedly abnormal ovaries compared to heterozygous and WT littermates. These ovaries were smaller and showed no signs of significant follicular growth or earlier ovulation (Fig. 2B, Supplementary Fig. 2A)—neither could be restored by hormonal stimulation with pregnant mare's serum gonadotropin (PMSG) and human chorionic gonadotropin (hCG).

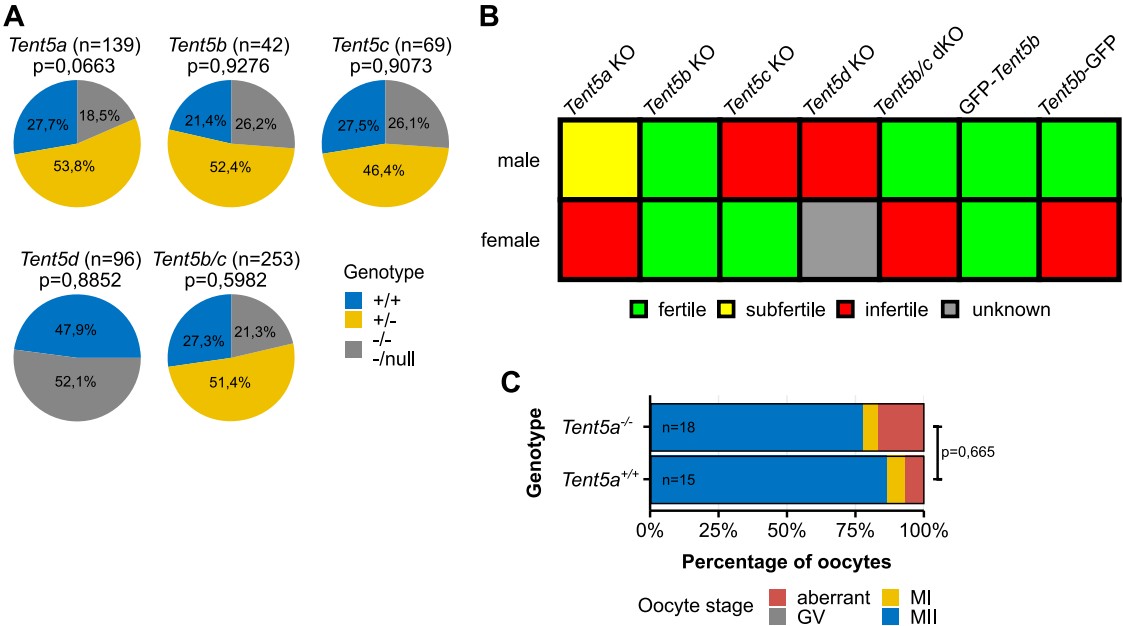

**Fig. 1 | TENT5 poly(A) polymerases are involved in both spermatogenesis and oogenesis. A** Distribution of genotypes in pups of all *Tent5* KO mouse lines from matings of heterozygous parents (only males were accounted for in the *Tent5d* KO line, in the *Tent5b/c* dKO mouse line, both parents had the *Tent5b⁻ Tent5c⁺/⁻* genotype, and for this, only *Tent5c* genotype distribution is presented). P values reported for comparison with expected Mendelian ratios in Fisher's exact test, two-tailed; n values represent the number of pups genotyped. **B** Fertility status of homozygous males and females of all *Tent5* KO and *Tent5b* GFP-tagged mouse lines. **C** In-vitro maturation efficiency of *Tent5a⁺/⁺* and *Tent5a⁻/⁻* GV oocytes. P value reported for comparing the ratio of metaphase II oocytes to other oocytes after 24 h of in-vitro culture in Fisher's exact test, two-tailed; n values represent the number of oocytes analyzed; GV germinal vesicle, MI metaphase I, MII metaphase II. Source data are provided as a Source Data file.

**Table 1 | Summary of fertility phenotypes discovered for each knock-out and knock-in mutation of *Tent5* genes**

| | Male | Female |
|---|---|---|
| *Tent5a⁻/⁻* | Lowered fertility due to poor overall animal condition | Infertility due to poor overall animal condition |
| *Tent5b⁻/⁻* | Fertile | Fertile |
| *Tent5c⁻/⁻* | Loss of sperm head and DNA material; Fewer total germ cells produced | Fertile |
| *Tent5d⁻/null* or *Tent5d⁻/⁻* | Abnormalities in the development of primary spermatocytes; Degeneration of testes tissue; Lowered blood testosterone levels | Unknown—no homozygous females were analyzed due to *Tent5d* localization on chromosome X |
| *Tent5b⁻/⁻ Tent5c⁻/⁻* | Recapitulated Tent5c⁻/⁻ phenotype | Inhibited ovarian follicle growth; No ovulation; Degeneration of preovulatory prophase I oocytes |
| *Tent5b^gfp/gfp* (N-terminal) | Fertile | Fertile |
| *Tent5b^gfp/gfp* (C-terminal) | Fertile | Abnormalities in spindle and chromatin organization in metaphase II oocytes; No ovulation |

Dissection of these 8-week-old *Tent5b/c* dKO ovaries revealed that all oocytes were in various stages of degeneration and cell death. WT females of the same age maintained a significantly high (85%) population of live oocytes. In comparison, ovaries from single KO females (*Tent5b⁻/⁻* or *Tent5c⁻/⁻*) contained a pool of viable oocytes comparable to that of WT females, while females with a single WT allele of either gene (*Tent5b⁺/⁻c⁻/⁻*, *Tent5b⁻/⁻c⁺/⁻*) appeared to have only a slightly lower number of living oocytes. To verify whether this was a congenital or acquired condition, we also dissected 5-week-old *Tent5b/c* dKO ovaries. In these, oocytes showed initial signs of degeneration as seen in adult dKO females, with 35% of oocytes remaining in the living state and the remainder showing various abnormalities—a result that differed significantly from both WT and 8-week-old *Tent5b/c* dKO females (Fig. 2B, C).

Histological staining of individual sections of the ovaries confirmed the arrest of follicular growth in *Tent5b/c* dKO females at the early antral stage. We found neither mature antral follicles ready for ovulation nor the presence of corpora lutea, indicating previous ovulation (Fig. 2D). In *Tent5b/c* dKO females, the observed follicle diameter ranged from 39.8 to 227.8 μm, whereas in WT females, diameters ranged from 30.85 to 507.7 μm (Fig. 2E). Additionally, high levels of apoptosis detected with the TUNEL assay were exclusive to *Tent5b/c* dKO ovaries, observed not only within oocytes but also in surrounding granulosa cells. Ovaries of WT, *Tent5b* KO, and *Tent5c* KO females showed only minor traces of positive TUNEL assay staining (Fig. 2F, Supplementary Fig. 2B), completing the picture of the deterioration of entire *Tent5b/c* dKO ovaries—at morphological, physiological, and cellular levels.

## Dominant effect of TENT5B C-terminal GFP knock-in

A model that provided us with additional insight into the role of TENT5B and TENT5C in oocyte biology was a mouse knock-in line

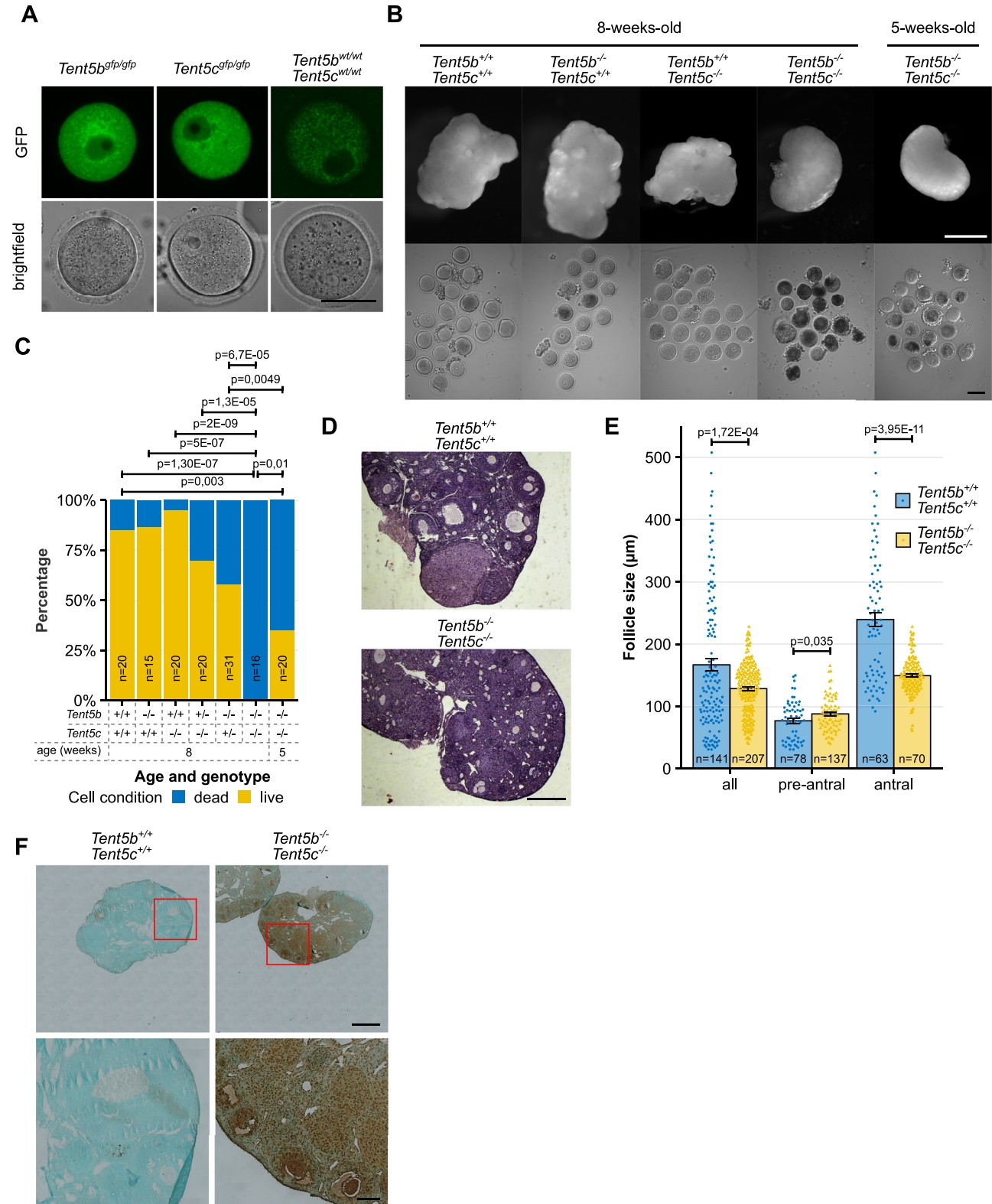

expressing TENT5B with a C-terminal GFP tag (TENT5B-GFP), which, as previously stated, displayed specific infertility in females. On the other hand, N-terminal tagging of TENT5B (GFP-TENT5B) or C-terminal tagging of TENT5C (TENT5C-GFP) did not lead to any oocyte abnormalities. This contrast accompanied a difference in the expression of TENT5B-GFP depending on the GFP fusion site—the mean expression of TENT5B-GFP, measured as the level of GFP fluorescence intensity, was over three times higher than that of GFP-TENT5B (Fig. 3A, B).

The detrimental effect of GFP expression at the C-terminus of TENT5B manifests as an extreme drop in litter sizes, the severity of which depended on the number of GFP alleles in females, thus pointing to a potential dominant effect. To accurately assess this effect, we set up a series of matings with different female and male genotype configurations. We monitored them for 10 weeks and compared the average number of pups birthed by females per week (for the total observation period of 10 weeks). We observed that the negative effect

**Fig. 2 | Lack of TENT5B and TENT5C leads to early oogenesis arrest. A** Expression pattern of TENT5B-GFP and TENT5C-GFP in GV oocytes visible as fluorescence of GFP. Minor fluorescence in WT oocytes represents background autofluorescence in GFP channel. Scale bar = 50 μm. **B** Morphology of ovaries and GV oocytes isolated from adult (8-week-old) and young (5-week-old) female mice. Top scale bar = 1 mm, bottom scale bar = 100 μm. **C** Percentage of live and dead GV oocytes isolated from the ovaries of female mice carrying different combinations of *Tent5b* and *Tent5c* genotypes. N values represent number of oocytes counted; p values reported for comparison of the ratio of live to dead oocytes in Fisher's exact test, two-tailed. **D** Haematoxylin and eosin staining of histological sections from *Tent5b⁺/⁺ Tent5c⁺/⁺* and *Tent5b⁻/⁻ Tent5c⁻/⁻* ovaries. Scale bar = 500 μm. **E** Distribution of ovarian follicle size in *Tent5b⁺/⁺ Tent5c⁺/⁺* and *Tent5b⁻/⁻ Tent5c⁻/⁻* females, with the analysis of all oocytes further broken down into pre-antral and antral follicle stages. Size value represents the mean width of the follicle at the widest point and width perpendicular to that measurement at the narrowest point, measured on each follicle's widest cross-section available. Individual data points and n values represent individual follicles measured; bars represent mean value; error bars represent SEM; p values reported for follicle size comparison in the t-test, two-tailed. **F** TUNEL assay staining for signs of apoptosis in histological sections from *Tent5b⁺/⁺ Tent5c⁺/⁺* and *Tent5b⁻/⁻ Tent5c⁻/⁻* ovaries. Upper scale bar = 500 μm, lower scale bar = 100 μm. See also Supplementary Fig. 2B. Source data are provided as a Source Data file.

of C-terminal GFP on fertility was limited to females: *Tent5b^{wt/gfp}* and *Tent5b^{gfp/gfp}* females suffered from a significant drop in fertility, with a much stronger effect observed in homozygotes (Fig. 3C). Meanwhile, both *Tent5b^{wt/wt}* and *Tent5b^{gfp/gfp}* males parented a similar number of pups when mated with WT females.

Simultaneously, we searched for the cause of this lowered fertility. Knowing the phenotype of *Tent5b/c* dKO females, we focused our investigation on oocyte development. Unlike *Tent5b/c* dKO, oocytes from *Tent5b^{GFP}* (both *Tent5b^{wt/gfp}* and *Tent5b^{gfp/gfp}*) females remained alive in the ovaries and progressed further in meiosis through GV to the second metaphase (MII) stage. However, chromosome spreads from these MII oocytes isolated from heterozygous (at the time, homozygous *Tent5b^{gfp/gfp}* females were born at an extremely low rate and were unavailable for these experiments) and control WT females revealed the first signs of the cause underlying observed infertility (Fig. 3D). Although all oocytes of both groups maintained a standard number of 20 pairs of unaltered chromosomes, in most knock-in oocytes analyzed, either one or several chromosomes were separated from the main group of chromosomes, while all chromosomes were gathered in WT oocytes (Fig. 3E). Immunofluorescence staining of chromatin and spindle microtubules in MII *Tent5b^{wt/gfp}* oocytes, of which one-fourth suffered some disruption, indeed revealed disorganized spindles and chromosomes. These stainings additionally revealed parthenogenetic activation as well as large, seemingly empty or fluid-filled follicles within the cytosol (compared to only 2.38% in WT oocytes; Fig. 3F, G).

Together with the observation that the *Tent5b* KO mutation alone does not impact female fertility in any way, our results strongly suggest the dominant-negative or gain-of-function effect of the GFP tag when located at the C-terminus of endogenous TENT5B, as well as the importance of TENT5B activity in the oocyte.

## TENT5s polyadenylate mRNAs in oocytes, the tight regulation of which is essential for oogenesis

As described above, TENT5B and TENT5C play essential roles in oogenesis. To study the effect of dysfunction of these poly(A) polymerases at the mRNA level and identify their potential substrates, we performed direct RNA sequencing (DRS) using the nanopore-based MinION platform (Fig. 4A, Supplementary Data 1). DRS enables genome-wide poly(A) tail length profiling and does not suffer from PCR amplification-dependent biases, which can be particularly pronounced for long poly(A) tails often present in developing gametes. The limitation of this sequencing methodology, however, is that it requires a relatively large quantity of material, rendering the sequencing of RNA from groups of oocytes impossible in our setup. Therefore, we instead used total RNA isolated from whole ovaries of WT, *Tent5b/c* dKO, as well as homozygous and heterozygous TENT5B-GFP 30-day-old female mice.

*Tent5b/c* dKO had essentially no effect on the global mRNA poly(A) tail length distribution in ovaries (mean lengths: 103 nt in WT vs 106 nt in *Tent5b/c* dKO; Fig. 4B). However, analysis of the distribution of differential poly(A) tail lengths revealed in *Tent5b/c* dKO samples several mRNAs with significantly shorter tails, encoding proteins

involved in oogenesis, such as mRNA encoding Zona Pellucida Glycoprotein 3 (*Zp3*) or Growth Differentiation Factor 3 (*Gdf9*) (Fig. 4C, Supplementary Fig. 3, Supplementary Data 2). We further validated the poly(A) tail length distributions observed in DRS using a PCR poly(A) test (PAT; Fig. 4D).

For many of the mRNAs with shortened poly(A) tails in *Tent5b/c* dKO, we observed elongation in homo- and hetero-zygous *Tent5b^{GFP}* knock-in mice (Fig. 4C, Supplementary Fig. 3, Supplementary Data 3, 4). This is consistent with the elevated expression of TENT5B protein in the presence of a C-terminal GFP tag. In homo- and hetero-zygous *Tent5b^{GFP}* knock-in ovaries, we observed global shortening of poly(A) tails which most probably represents a secondary effect (Fig. 4B).

To specifically examine oocyte-enriched transcripts, we performed an RNA-seq analysis on oocytes isolated from 5-week-old WT and *Tent5b/c* dKO females using the Illumina platform (Fig. 4E). This allowed us to narrow down the number of transcripts to be analyzed to 522 highly expressed in oocytes (Supplementary Data 5), for which we analyzed the poly(A) tail lengths based on DRS data. This revealed 37 mRNAs that had shortened poly(A) tails in *Tent5b/c* dKO and 48 mRNAs with elongated tails in *Tent5b^{gfp/gfp}* (Fig. 4E). Notably, 20 mRNAs were common among these groups (Fig. 4F). From the pool of 75 potential TENT5 targets, we removed those that had shortened tails in *Tent5b^{gfp/gfp}* or lengthened tails in *Tent5b/c* dKO. We also removed targets that had elongated tails in *Tent5b^{gfp/gfp}* and slightly elongated tails in Tent5b/c dKO from the analyses. The combined poly(A) profile of all 55 mRNAs responding to TENT5 dysfunction (Fig. 4G) revealed an opposite effect of KO and TENT5B-GFP knock-in, supporting the hypothesis that GFP at the C-terminus of TENT5B leads to a gain-of-function phenotype. It further indicates that mRNAs with elongated tails in *Tent5b^{gfp/gfp}* or shortened tails in *Tent5b/c* dKO represent direct targets of TENT5s (Fig. 4E).

Interestingly, differential expression analysis performed on the WT and *Tent5b/c* dKO Illumina RNA-seq dataset indicated no downregulation of genes subjected to poly(A) tail shortening in *Tent5b/c* dKO. Nonetheless, immunohistochemistry staining in *Tent5b/c* dKO ovaries (no staining was performed on *Tent5b^{gfp/gfp}* ovaries due to the very low birth rate of those females) revealed a drastic reduction of GDF9 and, to a lesser extent, the level of ZP3 protein in later stages of follicle development. This is in accordance with the observed phenotype of follicle arrest in development at the early antral stage (Fig. 4H, Supplementary Fig. 4). Collectively, these observations point to the previously suggested dominant role of poly(A) tail dynamics in the regulation of translation during gametogenesis.

In summary, TENT5B and TENT5C polyadenylate a group of mRNAs essential for oogenesis, enhancing their expression. Notably, both the lack and excess of TENT5-mediated poly(A) tail extension lead to aberrant oogenesis.

## TENT5C and TENT5D are essential for different stages of spermatogenesis

While neither *Tent5* gene is single-handedly responsible for undisturbed oocyte development, such is not the case for spermatogenesis —as already mentioned, both *Tent5c* and *Tent5d* KO mutations render

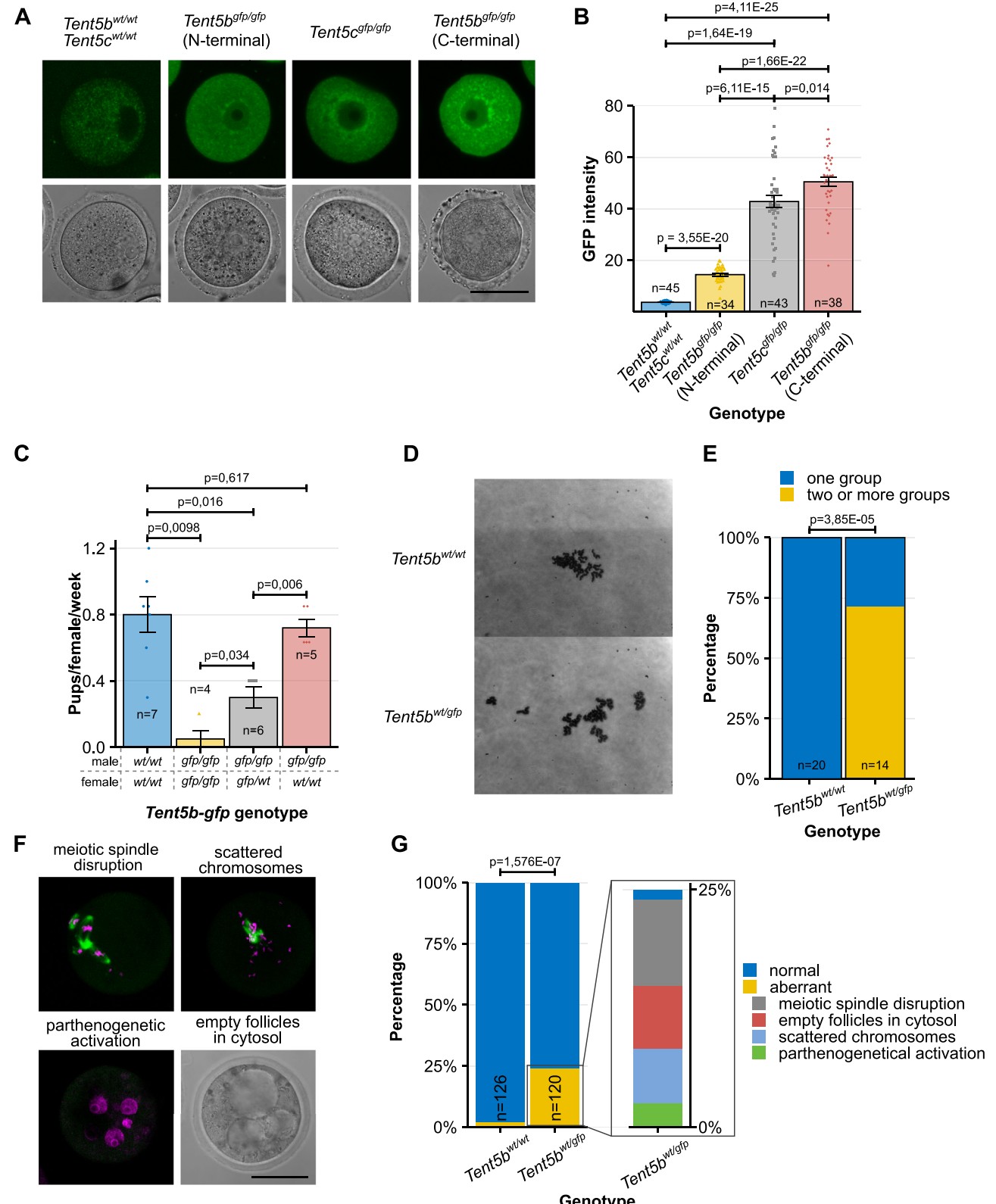

males sterile[31–34]. However, the expression of these two proteins is markedly different between testis and developing sperm. Immunofluorescence detection of TENT5C-GFP and TENT5D-GFP in histological sections of testis shows high expression of TENT5C close to the lumen of the seminiferous tubules in both round and elongated spermatids for 3- and 8-week-old males. At the same time, TENT5D displays a rather uniform expression across testis tissue in young

males, with limited amounts in earlier meiotic stages, and fades from the lumen in adult males (Fig. 5A).

To check the exact pattern of TENT5C and TENT5D expression, we sorted germ cells isolated from males with TENT5C and TENT5D GFP- and FLAG-tag knock-in into different stages of spermatogenesis based on DNA content and chromatin structure (Supplementary Fig. 5) and used them for western blot (FLAG tag lines) and flow cytometry (GFP

**Fig. 3 | Dominant effect of TENT5B C-terminal GFP knock-in. A** Expression pattern of TENT5B-GFP, TENT5C-GFP, and GFP-TENT5B in GV oocytes visible as fluorescence of GFP. Scale bar = 50 μm. **B** Comparison of TENT5B and TENT5C expression levels depending on the presence of the GFP tag at the N- or C-terminus of proteins, measured as a GFP fluorescence intensity. Individual data points and n values represent individual oocytes analyzed, bars represent mean intensity values, error bars represent SEM; p values reported for comparison of mean values in the t-test, two-tailed. **C** Birth rate for different mating configurations depending on male and female C-terminal *Tent5b^{GFP}* genotype, presented as the number of pups born by individual females per week. Individual data points and n values represent individual females observed for 10 weeks, bars represent mean value, error bars represent SEM; p values reported for comparison of the number of pups in the Mann–Whitney–Wilcoxon test, two-sided. **D, E** Chromosome scatter preparations showing chromosome conditions and organization in *Tent5b^{wt/wt}* and *Tent5b^{wt/gfp}* MII oocytes. N values represent the number of individual oocytes analyzed; p value reported for comparison of the ratio of oocytes with one group of chromosomes to oocytes with chromosomes separated into two or more groups in Fischer's exact test, two-tailed. **F** Chromatin (magenta) and alpha-tubulin (green) immunofluorescence staining in *Tent5b^{wt/gfp}* MII oocytes. Visible are examples of different chromosomal organization errors and other aberrations occurring after oocyte ovulation. Scale bar = 50 μm. **G** Frequency distribution of different chromosomal segregation and chromatin organization abnormalities in *Tent5b^{wt/gfp}* MII oocytes. N values represent individual oocytes analyzed; p value reported for comparison of the ratio of normal to abnormal oocytes in Fischer's exact test, two-tailed. Source data are provided as a Source Data file.

lines) analyses. Our results confirmed that TENT5C is initially expressed in secondary spermatocytes (spcII), with high protein levels maintained throughout the rest of the spermiogenesis process, while TENT5D expression, already present in primary spermatocytes, peaks in spcII cells to rapidly drop in round- and elongated spermatids (RS and ES, respectively) (Supplementary Fig. 6A–D).

Expression patterns of TENT5C and TENT5D correspond to sperm phenotypes that we and others[31–34] observed—*Tent5c* KO causes a high rate of head or DNA material loss in sperm isolated from the epididymis (Fig. 5B), while in *Tent5d* KO mice, testes become relatively smaller throughout the life of mice as compared to WT males (Supplementary Fig. 6E). This is a result of severe tissue structure deterioration, rendering the testes of 28-week-old males "hollow" (Fig. 5C). Consequently, blood testosterone levels in *Tent5d* KO males were significantly lower than in WT males (Supplementary Fig. 6F).

To gain deeper insight into the spermatogenesis process, we performed a more detailed flow cytometry analysis of the cell cycle. *Tent5c* KO males produce, on average, fewer total germ cells than WT males (Fig. 5D) but accumulate more ES in the lumen of seminiferous tubules (Supplementary Fig. 6G). *Tent5d* KO hinders germ cells from progressing through the meiosis – analysis of germ cell content in males aged 3, 4, 7, 10, and 15 weeks (single male per timepoint) showed that, while in WT males the population of primary spermatocytes (defined in flow cytometry by 4C DNA content) diminishes throughout mice life in favor of a rapidly growing number of spermatids and spermatozoa (collectively identified by 1C DNA content), in KO mice, this process is absent, and the ratio of germ cell populations remains stable through 15 weeks of mice life (Fig. 5E).

Then, we determined whether changes in the germline alone cause abnormal germ cell development and survivability or also depend on surrounding somatic cells. To this end, we co-cultured WT, *Tent5c* KO, and *Tent5d* KO germ- and Sertoli cells in different configurations for 48 h and accounted for live cells following that timepoint. For both genes, KO germ cells exhibited higher mortality than WT cells regardless of the presence of Sertoli cells in culture. Culturing germ cells with Sertoli cells, regardless of their genotype, improved germ cell survivability. This effect was independent of the genotype of Sertoli cells. Mortality and its improvement in an environment of Sertoli cells were particularly visible in the case of *Tent5d* KO (Fig. 5F) and to a lesser extent for *Tent5c* KO (Supplementary Fig. 6H).

All results described above, as well as the differential localization of TENT5s, indicate that the phenotypes are mainly related to germ cells and that the substrates for TENT5C and TENT5D in testes may differ. To identify these substrates, we have performed DRS of mRNA isolated from the testes of adult *Tent5c* KO and 3-week-old *Tent5d* KO males (due to their age-dependent testicular degeneration). *Tent5c* and *Tent5d* KO mutations both had only minor effects on the global poly(A) tail length profile (Fig. 5G, Supplementary Data 6, 7). Such effects can be partially attributed to changes in cell populations. At the same time, the age of mice has quite a dramatic effect on the global poly(A) tail length distribution, which reinforces the importance of comparing males at the same age (a more thorough study of the effect of age on the activity of TENT5 proteins and poly(A) tail dynamics was out of the scope of this work). Importantly, we identified potential TENT5 substrates with significantly shorter poly(A) tails, with no overlap between mRNAs identified in *Tent5c* and *Tent5d* KO samples (Fig. 5H, Supplementary Data 6, 7). Among mRNAs affected by TENT5 dysfunction in sperm, particular attention is owed to several that play important roles in gametogenesis: *Tppp2*, *Insl3* (in *Tent5c* KO), and *Rnaset2* (in *Tent5d* KO) (Fig. 5I–K). We validated the changes in poly(A) tail length distribution for *Tppp2* and *Insl3* in *Tent5c* KO germ cells and whole testes, respectively, by PAT assay (Supplementary Fig. 6I). Finally, quantitative immunochemistry analysis revealed that the shortening of their mRNA poly(A) tails leads to decreased expression at the protein level during stages of spermatogenesis corresponding to KO phenotypes: TPPP2 in *Tent5c* KO spermatids and RNASET2 in *Tent5d* KO spermatocytes. Additionally, INSL3 expression was lowered in *Tent5c* KO Leydig cells (Fig. 5L–N).

## TENT5s enhance the expression of secreted proteins during gametogenesis

Poly(A) tail profiling using DRS allowed us to identify substrates of TENT5 poly(A) polymerases in testes and ovaries, many of which play essential roles during gametogenesis. This raises the question of the mechanism leading to substrate specificity. To this end, we first searched for sequence features of mRNAs targeted by TENT5B and TENT5C in ovaries, TENT5D in young testes (due to strong deterioration of testicular tissue in adult males), and TENT5C in adult testes. Given that the best-described specificity factors for cytoplasmic polyadenylation are CPEB proteins, we screened the 3'UTR sequences of these transcripts identified by DRS for CPEB1 and CPEB2 motifs. However, no enrichment was found. Moreover, analysis of poly(A) tail length distribution of mRNAs possessing CPEB1/2 motifs showed essentially no effect of TENT5 KO (Fig. 6A, Supplementary Fig. 8A–D). Thus, it is unlikely that CPEB proteins play a role in TENT5-mediated cytoplasmic polyadenylation.

Then, we looked for differences in more general parameters of mRNAs regulated by TENT5s. We first performed 3'UTR and 5'UTR motif analyses on transcripts identified as TENT5 substrates. No specific motif was found except for the canonical polyadenylation signal (Supplementary Fig. 7A, B). Differences in the length of UTR segments, exon length, and GC content (Supplementary Fig. 7C, D), although statistically significant, were negligible.

As sequence analysis did not provide clues to the potential mechanism of substrate selection by TENT5s, we moved on to a functional analysis. Gene ontology (GO) analysis revealed genes involved in reproductive and developmental processes as the most enriched terms (Supplementary Fig. 7E). Moreover, a noticeable fraction of TENT5 substrates were mRNAs encoding secreted proteins (Fig. 6B), which also constituted the majority of TENT5 mRNA substrates in testes. The detection of the endoplasmic reticulum (ER)-

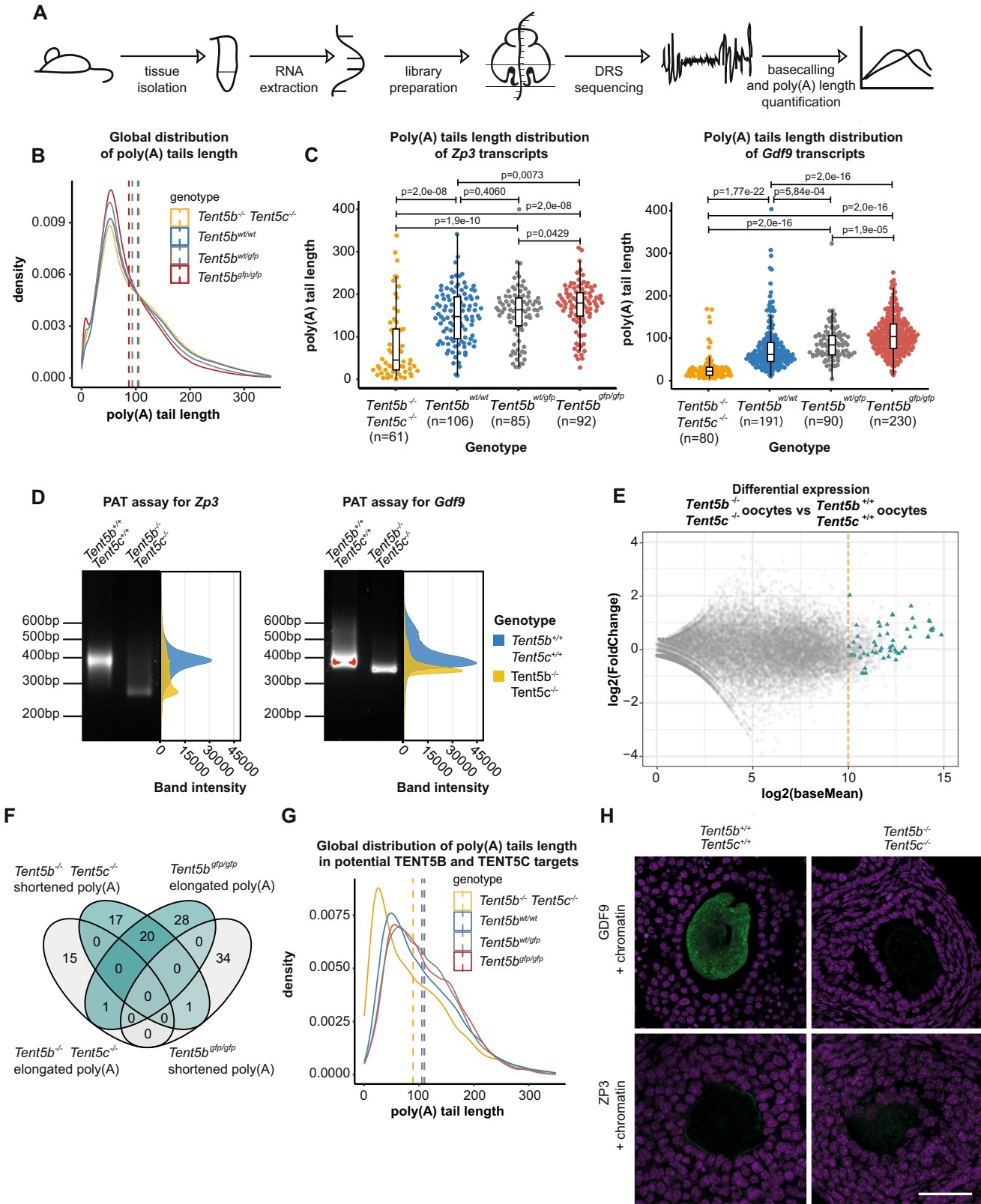

targeting signal peptide (SP) confirmed this view (Fig. 6C). SP was found in 6 out of 7 proteins encoded by TENT5D substrates (*Rnaset2a* included) and 8 out of 21 TENT5C substrates in testes. This fraction was smaller in ovaries, with only 9 out of 55 TENT5B/C substrates encoding proteins with SP. Notably, in both testes and ovaries, the effect of TENT5 dysfunction on the lengths of poly(A) tails was more dramatic for mRNAs encoding proteins with ER-targeting signal than for other

affected mRNAs, indicating that the former may represent major primary targets of TENT5s (Fig. 6C).

To investigate whether the presence of a SP is a determining factor for TENT5-mediated regulation in the oocyte, we injected GV oocytes with mRNA reporters coding, in 5′–3′ direction: SP (either *Gdf9*- or *Zp3*-specific), YPET fluorescent protein, short ER-retention KDEL motif. Each reporter also had a short oligo-A tail of 20-adenines

**Fig. 4 | TENT5s polyadenylates mRNAs in oocytes, the tight regulation of which is essential for oogenesis. A** Schematic of the DRS-based poly(A) tail length profiling. **B** The global distribution of poly(A) tail lengths of RNA isolated from the ovaries of *Tent5b⁻/⁻ Tent5c⁻/⁻*, *Tent5b^wt/wt*, *Tent5b^wt/gfp*, and *Tent5b^gfp/gfp* mice. Dashed lines indicate the mean poly(A) tail lengths: *Tent5b⁻/⁻ Tent5c⁻/⁻* = 106 nt; *Tent5b^wt/wt* = 103 nt; *Tent5b^wt/gfp* = 93 nt; *Tent5b^gfp/gfp* = 87 nt. **C** DRS-based poly(A) tail lengths profiling of *Zp3 and Gdf9* mRNAs isolated from ovaries. Median poly(A) tail lengths were marked as a horizontal line on boxplot, borders of box are IQR and black vertical lines correspond to 1.5 IQR. Median calculated for *Zp3*: *Tent5b⁻/⁻ Tent5c⁻/⁻* = 49 nt; Tent*5b^wt/wt* = 141 nt; Tent*5b^wt/gfp* = 156 nt; Tent*5b^gfp/gfp* = 170 nt; p values reported for multiple pairwise comparisons in the Mann–Whitney–Wilcoxon test, two-sided, with Bonferroni Hallberg correction. Median poly(A) tail lengths for *Gdf9*: *Tent5b⁻/⁻ Tent5c⁻/⁻* = 28 nt; Tent*5b^wt/wt* = 66 nt; Tent*5b^wt/gfp* = 78 nt; Tent*5b^gfp/gfp* = 106 nt; p values reported for multiple pairwise comparisons in the Mann–Whitney–Wilcoxon test with Bonferroni Hallberg correction; individual data points represent individual reads analyzed. **D** PAT assay visualizing the distribution of poly(A) tail lengths for *Zp3* and *Gdf9* transcripts from whole mRNA ovary-isolated samples. **E** RNA-seq performed on oocytes isolated from *Tent5b⁻/⁻ Tent5c⁻/⁻* and *Tent5b^wt/wt* mice. Yellow dashed line separates 522 genes enriched in oocytes (log₂fold change = 2 or more). Most potential TENT5B/C targets, highlighted by triangles, are highly expressed in *Tent5b⁻/⁻ Tent5c⁻/⁻* oocytes. Differential expression calculated using DESeq2. **F** Venn diagram illustrating the overlaps in sets of transcripts with statistically significant changes in the poly(A) tail length when compared with the Mann–Whitney–Wilcoxon test, two-sided, alpha = 0.05. Minimum poly(A) tail length difference is 10 adenosines. **G** Changes in the global distribution of poly(A) tail lengths for the group of transcripts selected as potential TENT5B and TENT5C targets. Dashed lines indicate the mean poly(A) tail lengths. **H** Immunohistochemistry staining of GDF9, ZP3 (green), and chromatin (magenta) in ovaries of *Tent5b⁻/⁻ Tent5c⁻/⁻* and *Tent5b⁺/⁺ Tent5c⁺/⁺* female mice. Scale bar = 50 μm. Source data are provided as a Source Data file.

at the 3′end to facilitate further polyadenylation in the cell. Such a reporter, when polyadenylated in the oocyte, would produce full YPET protein, of which fluorescence changes would inform us on the rate of translation. To draw conclusions on polyadenylation rate, we controlled the translation process by co-injecting individual oocytes with in-vitro polyadenylated, mCherry-coding mRNA (which would not require further polyadenylation to be translated). We then normalized YPET fluorescence intensity to mCherry intensity for every oocyte, tying information on fluorescence changes directly to the polyadenylation rate. For this experiment, we used *Tent5b⁻/⁻ Tent5c⁺/⁻* oocytes as they remain viable in ovaries but may already display changes at the transcriptomic level.

We observed that when reporter mRNAs encoded SP that recruits them to the ER, their polyadenylation rate was significantly lower in *Tent5b⁻/⁻ Tent5c⁺/⁻* oocytes as compared to WT ones. We observed no such difference for reporter mRNAs encoding only YPET (without ER-targeting signal). Our results suggest that only those transcripts recruited to the ER upon translation initiation depend on further polyadenylation by TENT5 proteins (Fig. 6D).

## Discussion

In this study, we demonstrate the essential roles of TENT5B, C, and D poly(A) polymerases during gametogenesis in mice (Fig. 7A, B). In spermatogenesis, in addition to TENT5C and TENT5D, previous studies identified TPAP as a testis-specific poly(A) polymerase, which is present in the cytoplasm of spermatogenic cells; its knockout leads to male infertility[35]. Three distinct cytoplasmic poly(A) polymerases are thus essential for proper spermatogenesis. In contrast, there is redundancy between TENT5B and TENT5C in oogenesis, targeting overlapping pools of transcripts. It is known for years that maintaining the poly(A) tail balance in transcripts accumulated in the oocyte's cytoplasm during its development is crucial for further maturation and fertilization. Until the discovery of TENT5s, the main known mechanism postulated for mRNA poly(A) tail regulation in oocytes was the CPEB protein accompanied by a complex of proteins organized around it[22]. CPEB recognizes specific motifs in the mRNA sequence and was postulated to recruit multiple different effectors like PARN deadenylase and the GLD2 poly(A) polymerase to maintain a short oligo(A) tail of the transcript. When hormonal stimuli arrive to the oocyte, promoting its maturation, PARN is supposed to dissociate from the complex, allowing GLD2 to elongate the transcript's poly(A) tail, thus activating it. The unchanged fertility of *Gld2* KO mice undermines this model and may suggest that TENT5s act redundantly with GLD2[17]. However, the data we provide indicate that TENT5 substrates are largely devoid of CPEB-binding motifs. Additional TENT5B/C act at the stage of oocyte growth when transcription is active rather than during oocyte activation. Thus, the poly(A) polymerase or polymerases responsible for waves of cytoplasmic polyadenylation during oocyte activation remain to be established. These

findings reveal a previously unexpected complexity in the regulation of poly(A) tail dynamics during mammalian gametogenesis.

Among TENT5B and TENT5C substrates, there are mRNAs encoding proteins essential for oogenesis. These are oocyte-specific zona pellucida component (ZP3) and signaling protein GDF9. While early studies of murine *Zp3* KO and null mutations showed that it is essential for zona pellucida (ZP) formation and oocyte ovulation along cumulus complexes[36,37], recently described *Zp3* mutations in human patients led to empty follicle syndrome (EFS), where no oocytes could be retrieved from patients' ovarian follicles following hormonal stimulation. This was due to the degeneration of oocytes or their failure to develop properly[38,39]. GDF9 is expressed exclusively in the oocyte[40], but its role extends beyond it, affecting the proliferation of granulosa cells[41,42] and regulating the development of the entire ovarian follicle[43,44], with its deficiency phenotype resembling our observation in *Tent5b/c* dKO female mice. While *Tent5b* KO leads to the shortening of the poly(A) tails of its substrates, the addition of a GFP-tag at the C-terminus of TENT5B leads to the elongation of poly(A) tails, clearly indicating a gain-of-function, possible due to accumulation at the protein level. Interestingly, the unfolded C-terminal region of TENT5B has predicted degron sequences not present in its paralogues (Supplementary Fig. 9)[45]. We thus hypothesize that addition of a GFP-tag at the C-terminus, but not the N-terminus, may inhibit ubiquitination, thereby increasing the stability of TENT5B. From a biological point of view, it is not straightforward to explain the infertility phenotype associated with TENT5B-GFP, manifested by chromosome disorganization during MII, However, at the molecular level, TENT5B-GFP affects proteins that are secreted and appear, at first sight, not to participate in meiosis itself. Further research is needed to elucidate the exact reason for infertility caused by TENT5B gain-of-function. TENT5B is not highly expressed in testes; thus explaining the lack of its effect on spermatogenesis.

The roles of TENT5C and TENT5D in the testis have been previously described by others[31–34]. Yet, the exact reasons for the observed phenotypes remain to be established. These polymerases clearly affect the expression of proteins involved in spermatogenesis. TENT5C regulates the proper expression of TPPP2, a protein vital for sperm motility and its fertilization capacity, and INSL3, a gonad tissue-specific hormone expressed in Leydig cells[46], where it is indispensable for testicular descent[47,48] and promotes germ cell survival[49]. TENT5D controls the expression of extracellular ribonuclease RNASET2 involved in the innate immune response[50], tumor suppression[51], and most importantly for this work, the regulation of sperm motility[52,53]. *Rnaset2* KO mice were shown to recapitulate neurological degeneration symptoms of human RNASET2 deficiency[54]. Other mRNAs affected by TENT5D identified by us and others, such as *Clu, Cst9, Cst12*[32], presumably also contribute to the observed phenotypes. Notably, disruption of TENT5D leads to oligoasthenoteratozoospermia and male infertility in humans[31–34].

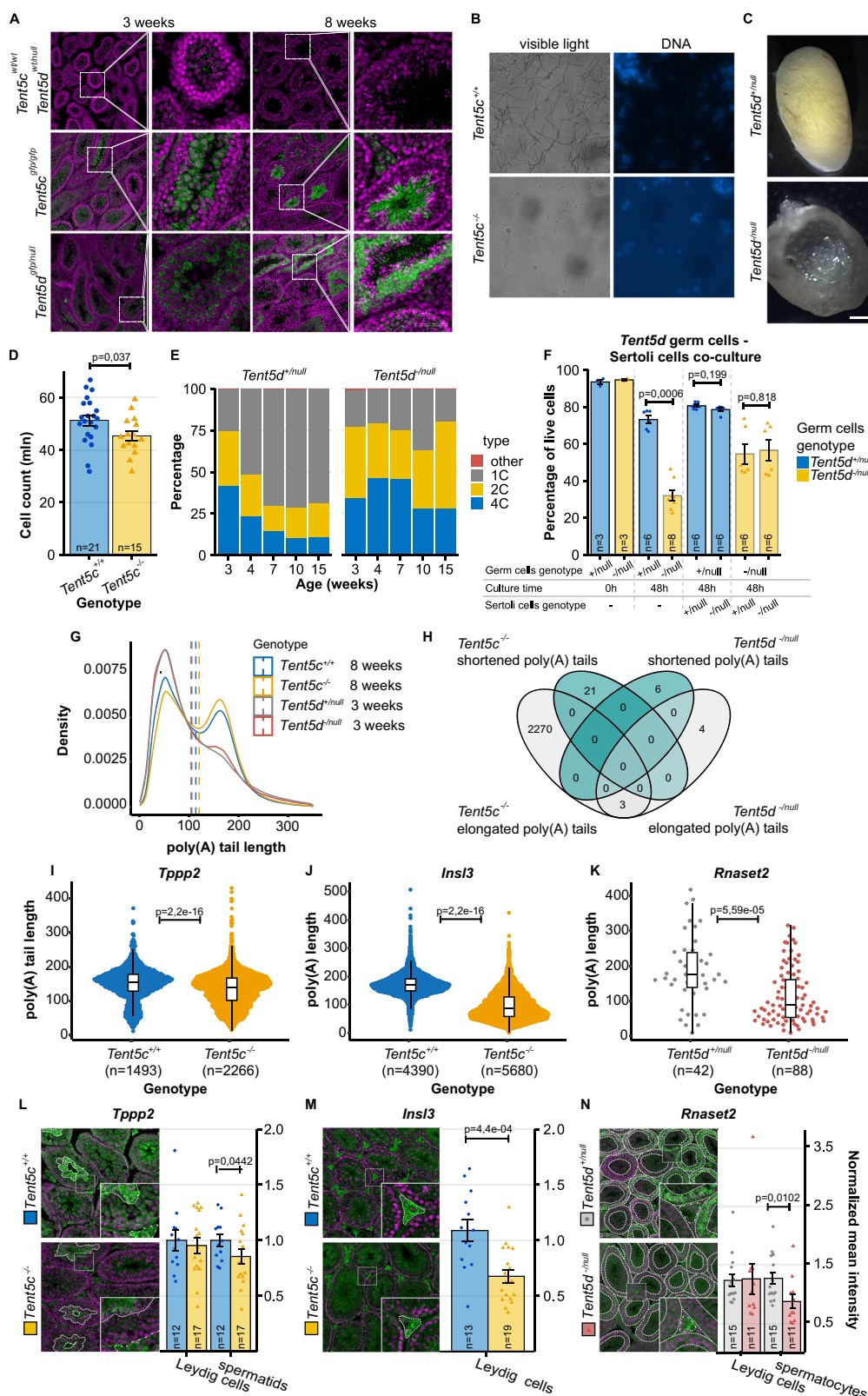

The mechanism of substrate recognition by TENT5s is far from being understood. However, it becomes apparent that in every biological context, these enzymes are enriched at the ER and, preferably, polyadenylate mRNAs that are translated at the ER (Fig. 7C). Indeed, previous studies identified immunoglobulin-encoding mRNAs as the most affected by *Tent5c* KO. TENT5A in osteoblasts[28] polyadenylates mRNAs encoding constituents of the extracellular matrix essential for bone mineralization, while in macrophages, substrates encode innate immune effectors. Such functions are conserved. In the worm *C. elegans*[30], the only TENT5 enhances the expression of secreted antimicrobial peptides in the gut. Here, we have shown that TENT5B, C, and D proteins are expressed in testes and oocytes, facilitating the expression of secreted proteins. Additionally, the various mRNAs regulated by TENT5s do not display specific sequence motifs. Notably,

**Fig. 5 | TENT5C and TENT5D are essential for different stages of spermatogenesis. A** Immunohistochemistry staining of TENT5C-GFP, TENT5D-GFP (green), and chromatin (magenta) in 3- and 8-week-old testes. Scale bar = 100 μm. **B** DNA staining of *Tent5c*$^{+/+}$ and *Tent5c*$^{-/-}$ sperm isolated from epididymis. **C** Cross section of 28-week-old *Tent5d*$^{+/null}$ and *Tent5d*$^{-/null}$ testes. **D** Total count of germ cells isolated from *Tent5c*$^{+/+}$ and *Tent5c*$^{-/-}$ testes. Individual data points and n values represent individual males used, bars represent mean value and error bars represent SEM; p value reported for t-test, two-tailed. **E** Changes in the distribution of germ cells with different DNA content (representing different stages of spermatogenesis) in *Tent5d*$^{+/null}$ and *Tent5d*$^{-/null}$ testes related to male's age (single male was used for each timepoint). **F** Germ cell survival at 48 h of in-vitro culture depending on the *Tent5d* genotype of germ cells and presence and *Tent5d* genotype of Sertoli cells. Individual data points and n values represent cultures of germ cells isolated from a single male, bars represent mean values, error bars represent SEM; p values reported for comparison in the Mann–Whitney–Wilcoxon test, two-sided. **G** The global distribution of poly(A) tail lengths of RNA isolated from *Tent5c*$^{+/+}$, *Tent5c*$^{-/-}$, *Tent5d*$^{+/null}$, *Tent5d*$^{-/null}$ mice. Dashed lines indicate the mean poly(A) tail lengths: *Tent5c*$^{+/+}$ = 113 nt; *Tent5c*$^{-/-}$ = 123 nt; *Tent5d*$^{+/null}$ = 103 nt; *Tent5d*$^{-/null}$ = 105 nt. **H** Venn diagram illustrating the overlaps in sets of transcripts with statistically significant changes in length of the poly(A) tails, Mann–Whitney–Wilcoxon test, two-sided, alpha = 0.05, minimum poly(A) tail length difference = 10 adenosines. **I, J** Changes in lengths of the poly(A) tails in transcripts essential for spermatogenesis. DRS-based poly(A) tail lengths profiling of *Tppp2* and *Insl3* mRNAs isolated from *Tent5c*$^{+/+}$ and *Tent5c*$^{-/-}$ testes. Median poly(A) tail lengths were marked as a horizontal line on boxplot, borders of box are IQR and black vertical lines correspond to 1.5 IQR. Median calculated for *Tppp2*: *Tent5c*$^{+/+}$ = 152 nt; *Tent5c*$^{-/-}$ = 136 nt; individual data points represent reads analyzed (partially obstructed due to the number of reads and resulting density of points); median poly(A) tail lengths for *Insl3*: *Tent5c*$^{+/+}$ = 169 nt; *Tent5c*$^{-/-}$ = 86 nt; p values reported for comparison in the Mann–Whitney–Wilcoxon test, two-sided. **K** Changes in poly(A) tail lengths in transcripts essential for the development of male gametes. DRS-based poly(A) lengths profiling of *Rnaset2* mRNAs isolated from *Tent5d*$^{+/+}$ and *Tent5d*$^{-/-}$ testes. Median poly(A) tail lengths were marked as a horizontal line on boxplot, borders of box are IQR and black vertical lines correspond to 1.5 IQR. Median calculated for *Rnaset2*: *Tent5d*$^{+/null}$ = 175 nt; *Tent5d*$^{-/null}$ = 88 nt; individual data points represent reads analyzed; p values reported for comparison in the Mann–Whitney–Wilcoxon test, two-sided. Immunohistochemistry staining in cross-sections of testes for chromatin (magenta) and proteins (green) encoded by transcripts identified as possible substrates of TENT5C (**L, M**) and TENT5D (**N**). Individual data points and n values represent mean values of all ROIs from individual specimens, bars represent mean values, error bars represent SEM, p value reported for the Mann–Whitney–Wilcoxon test, two-sided. Source data are provided as a Source Data file.

the incorporation of ER-targeting leader peptides into YFP-encoding mRNA leads to enhanced expression in WT oocytes but not in oocytes with TENT5B/C dysfunction (Figs. 6D and 7C). Accordingly, our recent data unexpectedly indicates that the Moderna vaccine mRNA-1273, which encodes Spike antigen targeted to the ER, is an efficient substrate for TENT5 polymerases. The Moderna vaccine is completely synthetic and optimized for efficient translation rather than cytoplasmic polyadenylation[55]. This points to a simple model in which efficiently translated mRNA with a leader peptide that are recruited to the ER are polyadenylated by TENT5s, which are enriched at the surface of the ER[29,30,56]. Further research is needed to elucidate how TENT5 poly(A) polymerases are recruited. Yet, the interaction of TENT5s with FNDC3A/B, membrane-bound proteins facing the cytoplasmic site of the ER, suggest that this is the most likely mechanism of such preference[56]. As TENT5s are differentially expressed, polyadenylation occurs only in specific cell types and tissues. For instance, many commonly used cell lines, such as HeLa or HEK293 cells, do not express TENT5s. Interestingly, there are also differences in the effect of TENT5-mediated polyadenylation between somatic cells and during gametogenesis. TENT5 proteins in B cells, macrophages, and osteoblasts mainly regulate mRNA stability. Dysfunction in these proteins thus leads to decreased levels of substrate mRNAs through the regulation of mRNA half-life. In contrast, during gametogenesis, TENT5 KO leads to the shortening of its substrates' poly(A) tails, leading to decreased levels of proteins produced, while mRNA levels remain unchanged. This agrees with previously proposed differences in poly(A) tail metabolism in somatic cells as compared to gametes and early embryos[57], with. In the latter, there are limiting amounts of the poly(A) binding proteins essential for efficient translation. As a result, poly(A) tail extension enhances protein synthesis at the same time in oocytes, and during spermatogenesis, mRNA stability is regulated differently than in somatic cells.

## Methods

### Materials and reagents availability
Plasmids, mouse lines, and reagents generated in this study are available at request from corresponding author.

### Experimental animal models
All experiments were performed using mouse lines generated using CRISPR/Cas9 method in C57BL/6/Tar x CBA/Tar mixed background.

All animal experiments were approved by the Local Ethical Committees in Warsaw affiliated to the University of Warsaw, Faculty of Biology (approval numbers: 176/2026, 917/2019) and Warsaw University of Life Sciences, Faculty of Horticulture and Biotechnology (approval numbers: WAW2/049/2022) and were performed according to Polish Law (Act number 266/15.01.2015).

In this study, females of following genotype and age were used: *Tent5a*$^{-/-}$ (8–12 weeks), *Tent5b*$^{-/-}$*c*$^{+/-}$ (4–12 weeks), *Tent5b*$^{-/-}$*c*$^{+/-}$ (8–12 weeks), *Tent5b*$^{gfp/gfp}$ (C-terminal) (8–12 weeks), *Tent5b*$^{gfp/gfp}$ (N-terminal) (4–12 weeks), *Tent5b*$^{gfp/wt}$ (N-terminal) (4–12 weeks), *Tent5c*$^{gfp/gfp}$ (8–12 weeks), wild-type (4–12 weeks), and males of following genotype and age were used: *Tent5b*$^{gfp/gfp}$ (8–48 weeks), *Tent5c*$^{-/-}$ (8–48 weeks), *Tent5d*$^{null/-}$ (3–48 weeks), *Tent5c*$^{gfp/gfp}$ (8–48 weeks), *Tent5c*$^{flag/flag}$ (8–48 weeks), *Tent5d*$^{gfp/gfp}$ (3 weeks), *Tent5d*$^{flag/flag}$ (8–48 weeks), wild-type (3–48 weeks).

The mice were kept in conventional polystyrene cages in the animal facility of the Faculty of Biology, University of Warsaw. A 12/12 light cycle was maintained in the room, 15 air exchanges per hour. The relative humidity in the rooms was 55% ± 10%. The temperature in the rooms was 22 °C ± 2 °C. During all procedures and breeding the animals received rodent feed and ad libitum water.

Sex of animals and cells used was determined by the according fertility phenotype connected to particular mutations.

### Mouse line generation
Generation of *Tent5a* KO and *Tent5c* KO mouse lines was described previously[28,29]. Procedure for generation of *Tent5a* KO, *Tent5b* KO, *Tent5d* KO, GFP-*Tent5b*, *Tent5b*-GFP, *Tent5c*-GFP, *Tent5c*-FLAG, *Tent5d*-GFP, and *Tent5d*-FLAG mouse lines is described below. Chimeric sgRNAs were synthesized by T7 RNA polymerase in-vitro transcription and purified by PAGE. Cas9 mRNA was in-vitro transcribed with T7 RNA polymerase (produced inhouse) and subsequently polyadenylated using *E. coli* poly(A) polymerase (NEB, Cat# M0276L) and m7Gppp5′N Cap was added using Vaccinia Capping System (NEB, Cat# M0280S). For knock-in mouse lines, repair DNA template was purchased as double stranded gene fragments (GeneArt, Thermo Fischer Scientific). See Supplementary Table 1 for sgRNAs, repair templates, and genotyping oligonucleotides sequences.

Zygotes obtained from mated females were microinjected into the cytoplasm using Eppendorf 5242 microinjector (Eppendorf-Netheler-Hinz GmbH) and Eppendorf Femtotips II capillaries with the following CRISPR cocktail: Cas9 mRNA (25 ng/μl), sgRNA (15 ng/μl), and donor dsDNA (7.5 ng/μl). After overnight culture microinjected embryos at 2-cell stage were transferred into the oviducts of 0.5-day p.c. pseudo-pregnant females. Born pups were genotyped by PCR at

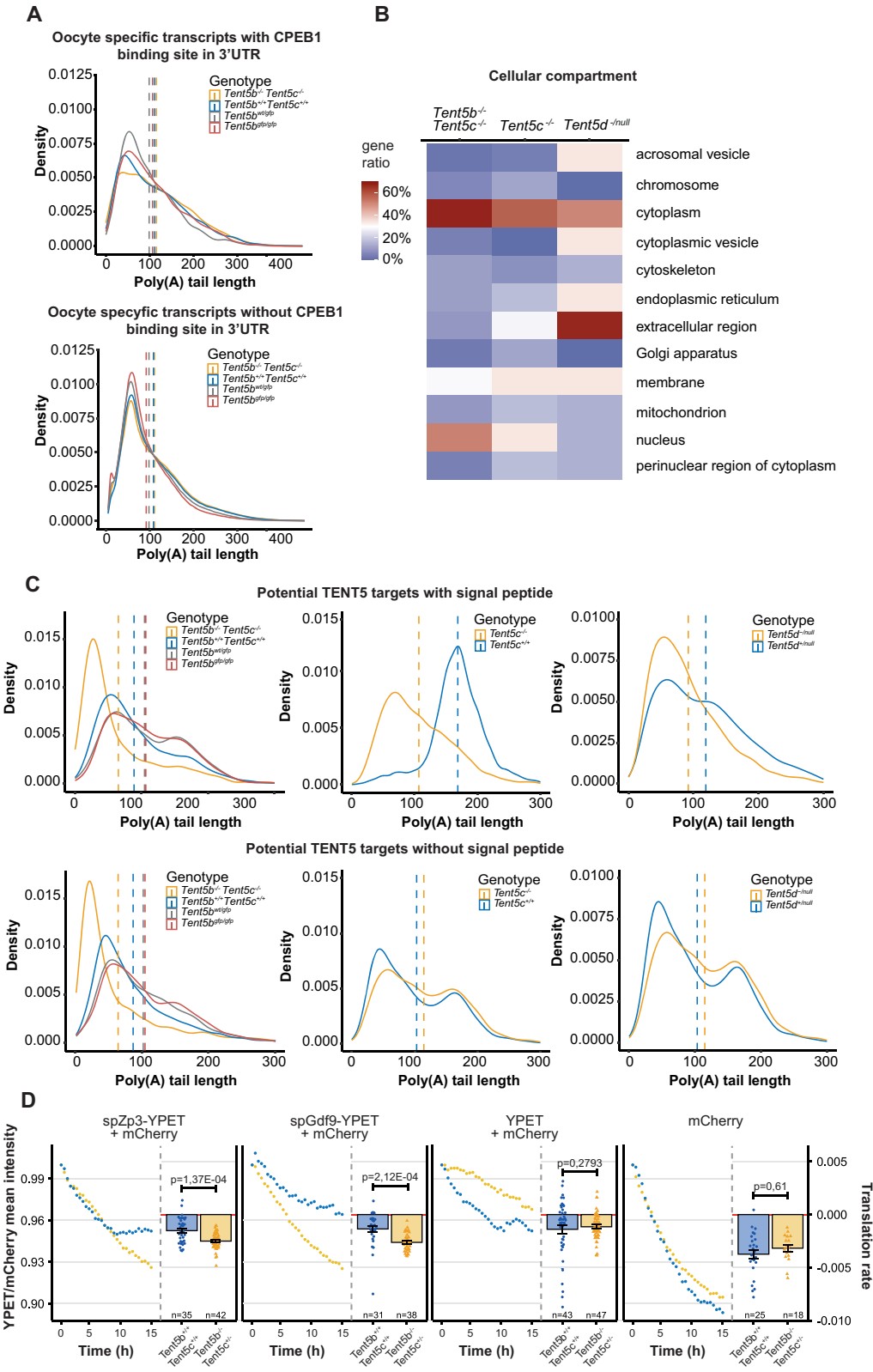

around 4 weeks. The presence of mutation was confirmed by sequencing in the founder mouse and, after backcrossing, in N1 generation mice. See Supplementary Information for primers' sequences used for genotyping.

**Fertility assessment**

Mice were deemed fertile when mating pair produced litters on regular basis within ~4 weeks intervals. Infertile mice never produced litter within span of at least 12 months. Every case of suspected infertility was confirmed by mating homozygous and heterozygous mice with wild-type mates.

**Cells and embryos isolation and culture**

To stimulate ovarian follicle growth and ovulation, adult female mice (aged 8–12 weeks) were injected intraperitoneally with pregnant mare's serum gonadotropin (PMSG) and human chorionic

**Fig. 6 | TENT5s enhance the expression of secreted proteins during gametogenesis. A** Differences in poly(A) tail profiles of transcripts containing CPEB1 motif in 3′UTR, compared to the rest of the transcriptome detected by DRS. In ovaries, analyses were performed for 522 oocyte-enriched mRNAs. Two-sided Mann–Whitney–Wilcoxon rank sum test was calculated. Dashed lines indicate the mean poly(A) tail lengths: $Tent5b^{-/-}$ $Tent5c^{-/-}$ = 115 nt; $Tent5b^{wt/wt}$ = 111 nt; $Tent5b^{wt/gfp}$ = 106 nt; $Tent5b^{gfp/gfp}$ = 99 nt for transcripts with CPEB1, and $Tent5b^{-/-}$ $Tent5c^{-/-}$ = 106 nt; $Tent5b^{wt/wt}$ = 104 nt; $Tent5b^{wt/gfp}$ = 94 nt; $Tent5b^{gfp/gfp}$ = 87 nt for transcripts without CPEB1 **B** Gene ontology (GO) analysis of potential TENT5 protein targets. **C** Distribution of poly(A) tail lengths in potential targets of TENT5 encoding proteins with and without signal peptide (SP). Dashed lines indicate the mean poly(A) tail lengths: $Tent5b^{-/-}$ $Tent5c^{-/-}$ = 65 nt; $Tent5b^{wt/wt}$ = 94 nt; $Tent5b^{wt/gfp}$ = 110 nt; $Tent5b^{gfp/gfp}$ = 109 nt; $Tent5c^{-/-}$ = 106 nt; $Tent5c^{+/+}$ = 168 nt; $Tent5d^{-/-}$ = 92 nt; $Tent5d^{+/+}$ = 119 nt for substrates with SP, and $Tent5b^{-/-}$ $Tent5c^{-/-}$ = 63 nt; $Tent5b^{wt/wt}$ = 85 nt; $Tent5b^{wt/gfp}$ = 101 nt; $Tent5b^{gfp/gfp}$ = 103 nt; $Tent5c^{-/-}$ = 103 nt; $Tent5c^{+/+}$ = 115 nt; $Tent5d^{-/-}$ = 100 nt; $Tent5d^{+/+}$ = 109 nt for substrates without SP. P values for TENT5B/C substrates with and without SP were calculated by two-sided Mann–Whitney–Wilcoxon test with Bonferroni Hallberg correction, and all p < 2.2e−12. P values for TENT5C and TENT5D substrates with and without SP were calculated by two-sided Mann–Whitney–Wilcoxon rank sum test, and all p < 2.2e−12 **D** Normalized mean intensity of fluorescence and calculated translation rates of oligoadenylated reporter YPET mRNAs with SP injected into $Tent5b^{+/+}$ $Tent5c^{+/+}$ and $Tent5b^{-/-}$ $Tent5c^{-/-}$ GV oocytes. Mean intensity data points represent mean YFP fluorescence intensity normalized to fluorescence of control polyadenylated reporter mRNA encoding mCherry. Individual translation rate data points and n values represent individual oocytes analyzed; bars represent mean values; error bars represent SEM; p values reported for the two-tailed t-test. Source data are provided as a Source Data file.

gonadotropin (hCG), respectively (10 units of hormone in 100 µl of PBS) in 44–46 h interval.

All procedures and culture of germinal vesicle (GV) and metaphase II (MII) stage oocytes as well as zygotes, unless stated otherwise, were performed in pre-warmed M2 or M16 medium (Sigma), additionally supplemented with 150 ng/µl dbcAMP (Sigma) to prevent meiosis resumption of GV prophase I-arrested oocytes when needed. For oocyte isolation, hormonally stimulated females were sacrificed either 48 h after PMSG stimulation for GV oocytes or 16 h after hCG stimulation for MII oocytes. For zygote isolation, hCG stimulated females were mated with males overnight and checked for presence of vaginal plug following day as the insemination indicator and then sacrificed 21–22 h post hCG injection. Ovaries and oviducts were dissected and immediately placed in M2 culture medium. For GV oocytes isolation, cumulus–oocyte complexes were released to medium by follicle puncture and cumulus cells surrounding oocytes were removed by pipetting. For MII oocytes and zygote isolation, oviducts were moved to dish containing warmed hyaluronidase solution (300 µg/ml in M2, Sigma) facilitating cumulus cells detachment, cumulus cells-surrounded oocytes/zygotes were released by ampulla puncturing and pipetted to disperse cumulus cells. Before and between further procedures, denuded oocytes were cultured in 10 µl drops of M2 medium under mineral oil in groups of 10–20 on plastic dishes (35 × 10 Tissue Culture Dishes, Falcon) placed in incubator at 37 °C and 5% $CO_2$ in the air.

For male germ cells isolation from testis, a protocol adapted from the work of Bastos et al.[58] was used. Dissected testes were decapsulated and placed in Falcon with 25 ml of HSBB buffer (Gibco) at room temperature. Next, 1 ml of type XI collagenase (12.5 mg/ml) was added (Sigma) and tubes were put to shake at 32 °C for 20 min in the shaking water bath (120 osc/min). Tubes were manually agitated every 5 min to facilitate the dissociation of the tubules. The tubules were washed once in 25 ml 1× HSBB buffer and resuspended in 10 ml of the HSBB buffer with 200 µl of the stock trypsin (25 mg/ml, Sigma) and 2 µl of the stock DNAse I (5 mg/ml, Sigma) and again put in a water bath with agitation at 32 °C for 15 min (120 osc/min). After the incubation, samples were dissociated for 4 min by pipetting 10 ml serological pipet and centrifuged at 450 × g for 3 min. The supernatant was discarded, and cells were resuspended in 1 ml of HSBB with 10% FBS (Gibco) and counted in Thoma chamber. Prepared cells were used for further procedures.

Isolation of Sertoli cells from testis was based on the protocol published by Bhushan et al. and Anway et al.[59,60]. Dissected testes were decapsulated and digested with 10 ml of trypsin (2.5 mg/ml)-DNase I (10 µg/ml) solution (Sigma) for 4 min in water bath at 32 °C (120 oscillations/min). Next, the trypsin digestion was stopped by adding 5 ml of trypsin inhibitor (10 mg/ml) (Sigma), samples were vigorously mixed and incubated for 5 min. After tubules settled down in the tube supernatant was carefully removed and 10 ml of trypsin inhibitor (2.5 mg/ml) was added. Digested tissue was washed nine times to remove all the germ cells. Remaining seminiferous tubules

were digested by collagenase (1 mg/ml)–hyaluronidase (1 mg/ml; Sigma)–DNase I (10 µg/ml) mixture followed by hyaluronidase (1 mg/ml)–DNase I (10 µg/ml) digestion. Finally, digested seminiferous tubules were passed ten times through the 18G needle. Isolated Sertoli cells were then seeded in the concentration of $1 \times 10^6$ cells/ml in RPMI medium (Gibco) supplemented in 10% FBS and cultured in 37 °C, 5% $CO_2$ until a monolayer of cells was obtained.

Previously isolated germ cells were seeded in the concentration of $2 \times 10^6$ cells/ml onto the Sertoli monolayer and cultured for 48 h. After 48 h medium containing germ cells were taken, and germ cells were washed with HSBB buffer and prepared for the flow cytometry analysis.

### Morphology, histology, cell visualization

For tissue histology analysis, H&E staining and TUNEL assay tissues were fixed in 10% neutral buffered formalin (Sigma) for 24 h, dehydrated in ethanol dilution series (70% ×1, 95% ×2, 100% ×3, changes every hour with overnight incubation in 4th change of 100% ethanol) and 2 changes of xylene, and incubated in two changes of liquid paraffin followed by paraffin embedding on metal trays using EC 350 Tissue Embedding Center (Myr). Paraffin-embedded tissues were sectioned using semi-automatic microtome (Leica RM2125 RTS) to obtain 10 µm-thick tissue slices transferred on Superfrost Ultra Plus glass slides (Thermo Fisher Scientific) and dried in 37 °C overnight before further processing.

For immunohistochemistry analysis, tissues were fixed in 4% paraformaldehyde (Sigma) in 0.1 M phosphate buffer for 24 h and cryopreserved in two changes of 30% sucrose (Merck) in 0.1 M PB, 24 h each. Tissues were frozen in Killik medium (Bio-Optica), cut into 20 µm-thick sections using cryostat and placed on Superfrost Ultra Plus glass slides. Sections were then stored in 4 °C until further processing.

Oocytes for immunofluorescence were fixed in 4% PFA (Thermo Fisher Scientific), permeabilized with 0.5% Triton X-100 (Sigma), 30 min in RT each, and blocked with 3% BSA (Sigma-Aldritch) in 4 °C overnight.

For nucleus–cytoplasm staining, paraffin-embedded sections on glass slides were first deparaffinized in two changes of xylene (10 min each) and rehydrated in 5 changes of ethanol (100% ×2, 96% × 2, 70% × 1, 2 min each) and distilled water for 2 min. Subsequently, slides were stained in Harris hematoxylin (Kolchem, Poland) for 10 min, rinsed for 2 min in tap water, dipped once in acidic ethanol (1% HCL solution in 70% ethanol), again briefly rinsed in tap water and for 2 min in tap water substitute (2.5 g NaHCO₃ and 20 g MgSo₄·7H₂O in 1 liter of water) and finally stained in eosin Y (Kolchem, Poland) for 1 min. After that, slides were dehydrated by reversing the rehydration protocol above (with 5 min xylene incubation instead of 10) and sealed using DPX mounting medium (Sigma) and a cover slide and dried overnight.

For apoptosis staining, tissue sections on glass slides were deparaffinized, rehydrated, and sealed as described above for H&E stained and a commercially available TUNEL Assay Kit (Abcam, Cat#

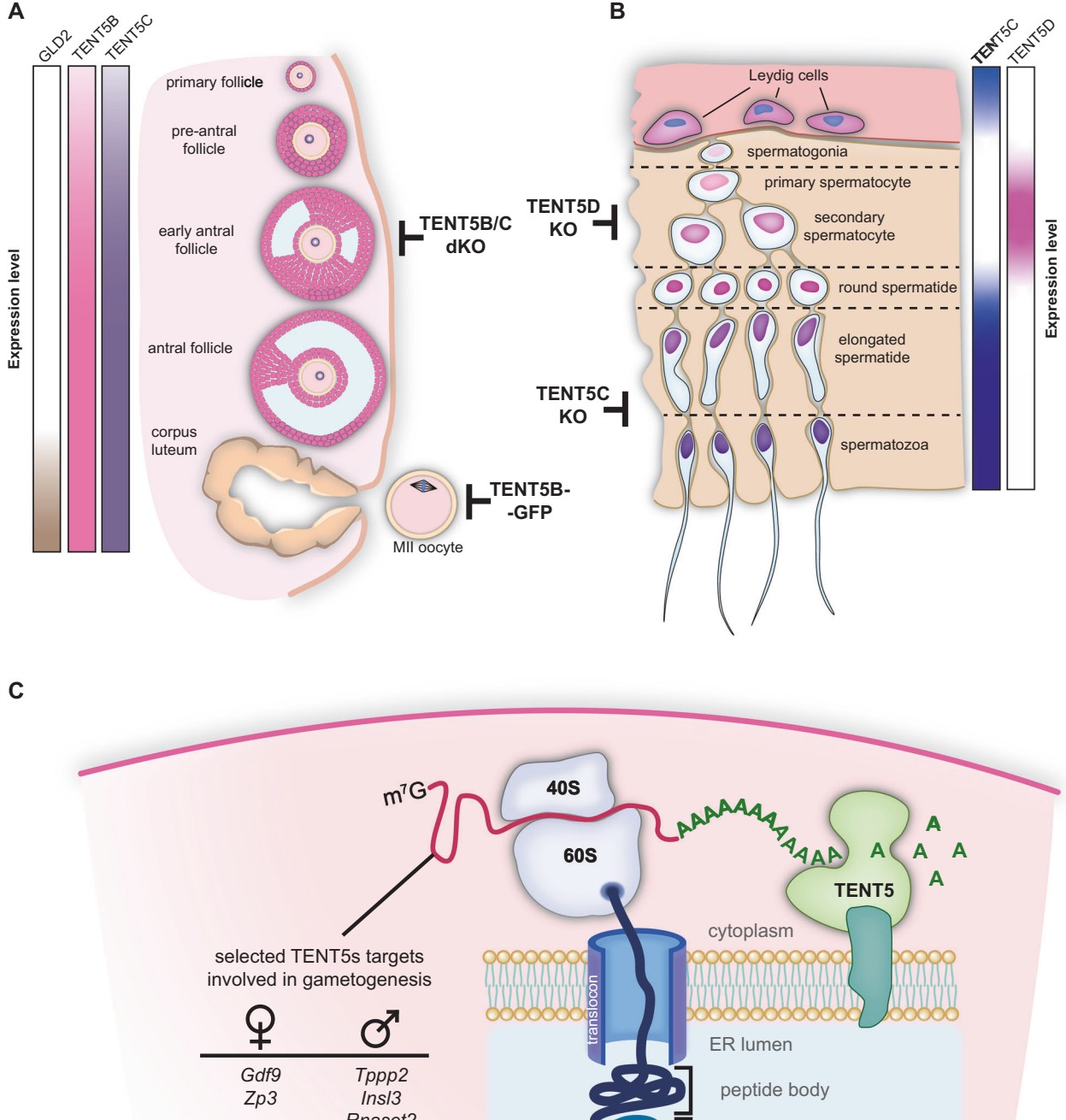

**Fig. 7 | Proposed model for the activity of TENT5 proteins in gametogenesis.** Simplified diagrams of oogenesis (**A**) and spermatogenesis (**B**). Black t-lines with labels indicate stages disrupted by a particular mutation. Gradient boxes indicate approximate expression of *Gld2* and *Tent5b, c,* and *d* genes at different stages of gametogenesis. **C** Proposed model for TENT5-mediated polyadenylation of transcripts encoding secreted proteins. After translation initiation, emerging signal peptide recruits the ribosome–mRNA complex to the ER, where further polyadenylation by TENT5 takes place.

ab206386) was used according to the manufacturer's protocol to detect double stranded breaks of DNA created during apoptosis. Negative control staining was prepared by substituting TdT in reaction mixture with water and positive staining control was prepared by treating sections with $1\,\mu g/\mu l$ DNase I in TBS (tris-buffered saline) for 20 min.

MII oocytes used for immunofluorescence were incubated overnight at 4 °C in primary monoclonal antibody against β-tubulin conjugated with FITC (Sigma-Aldritch Cat# F2043, Lot# 108M4766V; 1:50) and then washed twice in PBS for 15 min. To visualize chromatin, oocytes were stained in droplets of propidium iodide (0.01 mg/ml in PBS) at 37.5 °C, for 30 min on glass bottom dishes (MatTek Corp.).

To prepare chromosome spreads, MII oocytes were treated by Acidic Tyrode's Solution (Sigma) for about 30 s at 37 °C to remove zona pellucida. After wash in M2 medium oocytes were placed for 1–5 min in hypotonic solution of 1% sodium citrate. Next, single oocyte was placed on standard glass slide and fixed by several drops of methanol/acetic acid (3:1). Slides were air dried, and chromosomes were stained with 4% Giemsa solution for 10 min. Slides were washed in distilled water, air dried, washed in xylene, and finally mounted using DPX mounting medium.

For immunohistochemistry, frozen sections were dried for 10' at 55 °C, washed twice for 10' in TBS, incubated for epitope retrieval in pressure-cooker (70–90 kPa, 160 °C) for 2' in 20 mM Tris Buffer, pH 9.0 (for INSL3, TPPP2, and ZP3), pH 7.5 (for Rnaset2) or 10 mM citrate buffer, pH 6.0 (for Gdf9), and washed twice briefly with TBS. Next, sections were incubated 1 h at room temperature (RT) in blocking buffer (10% serum from the species that subsequently used secondary antibody was raised in, 1% BSA and 0.3% Triton X-100 diluted in TBS) and then overnight at 4 °C in moisture chamber with primary antibodies (anti-Tppp2 1:100, Abcam, Cat# ab236887; anti-INSL3 1:200, Invitrogen, Cat# PA5-55921 Lot: A96525; anti-RNASET2 1:100, Sigma-Aldrich, Cat# HPA029013, Lot: A115851; anti-GDF9, Abcam, Cat# ab254323, Lot: GR3279299; anti-ZP3, Proteintech, Cat# 21279-1-AP, Lot: 00041356) diluted in the same buffer. Next day slides were washed 3× for 5' with TBS-T (TBS with 0.025% Triton X-100), incubated 1 h at RT with secondary antibodies (donkey anti-rabbit Alexa 568-conjugated, 1:500, Invitrogen, Cat# A-10042, Lot: 2136776 or for GDF9 staining goat anti-rabbit Alexa 586-conjugated, 1:500, Invitrogen, Cat# A-11011, Lot: 2273773) diluted in blocking buffer with Hoechst 33342 (Sigma, Cat# H3570) 10 μg/ml, washed 3× for 5' with TBS-T and once with TBS.

To quench autofluorescence all sections except GDF9 staining were incubated with TrueBlack Plus(R) (Biotium) diluted 1× in PBS for 10', washed 3× for 5' with PBS. All sections were sealed with cover slide using ProLong Gold Antifade Mountant (Invitrogen).

## Flow cytometry analysis

In the seminiferous tubules of adult mammalians, germ cells in different maturation steps coexist, with 1C (round spermatids, elongating and elongated spermatids, spermatozoa), 2C (several types of G1 spermatogonia, secondary spermatocytes), and 4C (different stages of primary spermatocytes, G2 spermatogonia) DNA content[61]. For the efficient isolation of germ cells, we used protocol established by ref. 58. Considering the various sizes, shapes, and DNA content we employed fluorescence-activated cell sorting (FACS) as a convenient method to analyze cells at different stages of spermatogenesis, resulting in differentiating five populations for downstream analyses: gonial cells (gonia), several stages of primary spermatocytes (4C), secondary spermatocytes (spcII), round spermatids (RS), and elongated spermatids (ES).

For cell cycle analysis germ cells were stained with Vybrant™ DyeCycle™ Violet Stain (Thermo Fisher Scientific, Cat# V35003) 1 μl/ 1 mln of germ cells[62]. Cells were incubated at 32 °C in a water bath with agitation for 35 min (osc 90/min). Fluorescence was excited by 405 nm laser and DCV Blue fluorescence was detected with 450/50 filters while DCV Red fluorescence was detected with 525/50 filters. For the live/dead staining, the LIVE/DEAD™ Fixable Near-IR Dead Cell Stain Kit (Thermo Fisher Scientific, Cat# L34976) was used. Fluorescence of GFP was excited by 488 nm laser and detected with 530/30 filters. Samples were analyzed with BD LSRFortessa™ and sorted with BD Aria Fusion™ under FACS Diva Software v8.0.1 (BD) software control and analyzed using FlowJo (Data Analysis Software v10)[58,62].

## Imaging

Immunohistochemistry results were scanned as tile arrays on LSM800 (Zeiss) confocal microscope with 20× air objective. Fluorescence

intensities were extracted from images using Fiji (Is Just ImageJ)[63] [ImageJ ver. 1.53–1.54] with the help of Grid/Collection Stitching Plugin[64], manually indicated ROIs and self-written macros.

Oocyte live imaging was performed on Opera Phenix High-Content Screening System at 38 °C and 5% $CO_2$ in the air. Oocytes were placed in M16 medium on 384-well plate, one cell per well and were scanned for 15 h in 30 min intervals in brightfield, YFP and mCherry channels with five images gathered in z-axis with 40× water immersion objective every 10 μm with germinal vesicle center as a middle slice. Fluorescence intensity values were gathered from maximum intensity projection of all z-stack images for each oocyte using dedicated Opera Phenix software.

## Western blotting

For western blot analysis, whole tissue or equal amount of cells were lysed with 0.1% NP-40 in PBS supplemented with mix of protease inhibitors and viscolase (A&A Biotechnology) for 30 min at 37 °C with 600 rpm shaking, then Laemmli buffer was added and samples were denatured for 10 min in 100 °C. Samples were separated on 12–15% SDS–PAGE gels, proteins were transferred to Protran nitrocellulose membranes (GE Healthcare), and then membranes were stained with 0.3% w/v Ponceau S in 3% v/v acetic acid to control amount of the protein on the membrane for every sample. Membranes were then incubated with 5% milk or 5% BSA in TBST buffer according to the technical recommendations of the antibodies' suppliers for 1 h followed by incubation with specific primary antibodies diluted 1:1:000 (Flag Tag, Life Technologies, Cat# PA1-984B, Lot: WG319616) or 1:10,000 (GRP 94, Santa Cruz Biotechnology, Cat# sc-11402, Lot: C11616) overnight in 4 °C. Membranes were then washed three times in TBST buffer, incubated with HRP-conjugated secondary antibody: anti-goat (Millipore, Cat# 401393, Lot: 3924034) diluted 1:5000 for 1 h at RT. Membranes were washed three times in TBST buffer and proteins were visualized by enhanced chemiluminescence acquired on X-ray film (Fujifilm) using Clarity Western ECL Substrate (Bio-Rad).

## mRNA reporter oocyte microinjections

Original plasmids carrying mCherry- and YPET reporter-coding sequences, described before[65], were received on courtesy of prof. Marco Conti (UCSF). mCherry sequence-carrying plasmid was used unchanged to produce mCherry reporter mRNA. KDEL (along additional sequences; not used in this work) motif with STOP codon was cloned in ORF downstream of YPET-coding sequence with SLIC method using DNA fragments produced by PCR from mouse oocyte's cDNA (with forward primer containing KDEL motif-coding sequence)[66,67] and signal peptide-coding sequence of Zp3 or Gdf9 were inserted between YPET-coding sequence and T7 reverse transcriptase promoter in ORF by HindIII and BshTI digestion and ligation of oligonucleotides with sticky ends homologous to both enzymes' cutting sites. All plasmids' sequences were verified after cloning by Sanger sequencing. All oligonucleotide sequences are available in Supplementary Table 1.

To synthesize YPET-coding mRNA reporters from prepared plasmids, they were linearized by PCR using Phusion Hot Start II DNA polymerase (Thermo Fisher Scientific) with universal forward primer at YFP sequence and universal reverse primer in Plat 3′UTR sequence with 20 thymine residues overhang (see Supplementary Table 1. for primer sequences). For 50 μl reaction volume 10 ng of output DNA and 0.5 μM final primer concentration was used. The PCR reaction was run in thermal cycler with following program: 98 °C – 3 min, 30 cycles (98 °C – 10 s, 57 °C – 15 s, 72 °C – 30 s), 72 °C – 5 min, 4 °C – hold.

PCR product was run on agarose gel to verify obtained band size and purified using spin-column based Clean-Up kit (A&A Biotechnology).

Plasmid containing mCherry reporter was linearized by restriction digestion with MunI (MfeI) enzyme. For 40 μl of reaction 2.5 μl of DNA

sample was used. Digestion reaction was performed in 37 °C for 2 h and inactivated in 65 °C for 20 min. Digestion product was run on agarose gel to verify obtained band size and purified with 1 volume of AMPure XP (Beckman Coulter) beads and eluted with nuclease-free water.

Resulting DNA fragments where transcribed in-vitro using homemade batch of T7 RNA polymerase with following 40 μl reaction setup for 2 h in 37 °C: 12.5 ng/μl DNA template, 4 μl T-buffer, 10 mM $MgCl_2$, 2.5 mM of ATP, CTP, GTP, and UTP, 1.5 μl Ribolock, and 4 μl of T7 RNA polymerase, $H_2O$. mCherry reporter was additionally poly-adenylated in vitro by mixing whole 40 μl of in-vitro transcription reaction with 10 μl of poly(A) buffer (0.5 M Tris-HCl, 2.5 M NaCl), 10 μl 50 nM $MnCl_2$, 4 μl 100 mM ATP, 5 μl 5 U/μl *E.coli* poly(A) polymerase and 331 μl $H_2O$ and incubating reaction in 37 °C for 1 h.

All mRNA reporters were then cleaned by LiCl precipitation. Briefly, reactions were brought up to 400 μl volume, 200 μl of 7 M LiCl, 50 mM EDTA mix were added, and samples were incubated overnight at −20 °C. Then samples were centrifuged for 20 min at $16,000 \times g$, 4 °C, supernatant discarded, and precipitated mRNA washed with 1 ml of 70% ethanol. After centrifuging for 3 min at $14,000 \times g$ ethanol was discarded and air-dried pellet resuspended in RNase-free water.

Finally, all mRNA samples were capped using Vaccinia Capping System. 5 μg of mRNA in 30 μl was incubated for 5 min in 65 °C and 5 min in 4 °C before mixing with 4 μl of 10× capping buffer, 2 μl 10 mM GTP, 2 μl 2 mM SAM and 2 μl of Vaccinia Capping Enzyme and incubating for 30 min at 37 °C.

Capped mRNA was cleaned up by LiCl method as described above and resuspended in ~20–50 μl $H_2O$.

For injection, oocytes were isolated and allowed to recover for ~1.5 h. For microinjection, oocytes were transferred into new drops of pre-warmed medium. Microinjections were performed on Axio Observer 5 microscope (Zeiss) equipped with InjectMan 4, Transfer-Man 4r, CellTram 4r Oil, and FemtoJet 4i (Eppendorf). Mixture of 12.5 ng/μl mCherry mRNA, 12.5 ng/μl YPET reporter, and 0.05% NP-40 (NP-40 in PBS in nuclease-free water) in nuclease-free water was injected into cytoplasm with Femtotip II in minimal possible liquid volume (FemtoJet 4i set up for continuous leak, compensation pressure $p_c = 50$–150 hPa). After microinjection, oocytes were allowed to recover in culture for ~3 h before imaging.

## RNA isolation

Total RNA from oocytes was isolated using PicoPure RNA isolation kit (Thermo Fischer Scientific, Cat# KIT0204) according to manufacturer's protocol with following adjustments: groups of isolated oocytes were placed in 10 μl of provided extraction buffer in PCR tubes and incubated for 30 min at 42 °C in thermoblock. Samples were then stored in −80 °C. Before proceeding to next step, samples were pooled for total of 30 oocytes per sample and amount of provided ethanol used was adjusted depending on final sample volume. RNA was eluted with 11 μl of elution buffer provided. DNA was removed from final sample by DNase treatment using TURBO DNA-free Kit (Thermo Fisher Scientific, Cat# AM1907) according to manufacturer's protocol. DNA-free RNA was stored in −80 °C.

RNA from whole tissue samples and cultured cells was isolated using TRI Reagent (Sigma) using manufacturer's protocol. Tissues were homogenized in TRI Reagent using glass Dounce homogenizer and cell monolayers were rinsed with TRI Reagent and lysed cells were gathered for further isolation according to manufacturer's protocol. For all RNA samples DNA was removed as described for oocyte samples above.

## cDNA synthesis

cDNA was produced using SuperScript III reverse transcriptase (Thermo Fisher Scientific) according to manufacturer's protocol with oligo(dT) priming and addition of ERCC RNA Spike-InMix (final dilution: 1:100,000; Life Technologies) for first-strand synthesis from

oocyte-isolated RNA and RiboLock RNase inhibitor (40 units; Thermo Fischer Scientific) for second strand synthesis in all samples.

Output material was purified in 1 volume of AMPure XP beads, eluted in 20 μl nuclease-free water and second strand synthesis reaction was set up by addition of following components: Second Strand Buffer (NEB), RNase H (10 units; NEB), *E. coli* DNA ligase (10 units; NEB), *E. coli* DNA polymerase (50 units), dNTPs (final conc. 0.4 mM) and nuclease-free water for a final reaction volume of 50 μl. Reaction was incubated at 16 °C overnight. Final cDNA output was purified in 1 volume AMPure XP beads and eluted with 10 μl of nuclease-free water.

## RNA sequencing

Libraries from oocyte RNA for Illumina sequencing were prepared with application of tagmentation reaction, performed following published protocols[68,69] with various steps and amount of enzyme used optimized for use of homemade batch of Tn5 transposase produced in our laboratory. Briefly: 100 μM Tn5ME-A and Tn5ME-B oligonucleotides in annealing buffer (50 mM NaCl, 40 mM Tris-HCl pH = 8) were mixed 1:1 with 100 mM Tn5MErev oligonucleotide and incubated in thermocycler in 95 °C for 5 min and 65 °C for 5 min, and finally cooled to 4 °C and stored at −20 °C (with slow cooling between each step). Tn5 (0.25 mg/ml) was loaded with linker oligonucleotides Tn5ME-A/Tn5MErev and Tn5ME-B/Tn5MErev (see Supplementary Information file for full sequences) by mixing of 10 μl Tn5 with 0.5 μl of both linkers (0.35 μM) and incubating for 45 min in 23 °C with shaking at 350 rpm. Right before the reaction setup loaded Tn5 was diluted 10 times with nuclease-free water. 10 μl of freshly prepared tagmentation buffer (20 mM Tris-HCl pH 7.5; 20 mM $MgCl_2$; 50% dimethylformamide) were mixed with 5 μl of diluted Tn5 and 5 μl od cDNA, incubated for 3 min at 55 °C in preheated thermocycler and cooled to 10 °C for 1 min. Reaction was inactivated by addition of 5 μl 0.2% SDS and incubation for 5 min at RT. Tagmented cDNA was purified in 1.25 volume of AMPure XP beads and eluted in 10 μl of nuclease-free water. For library amplification KAPA HiFi HotStart ReadyMix 2x (Roche, Cat# KK2602) with addition of 5% DMSO was used. 5 μl of tagmented cDNA was used for 15 μl reaction volume. Reaction was run in thermocycler with following program: 72 °C – 15 min, 95 °C – 30 s, 15 cycles: (98 °C – 20 s, 58 °C – 15 s, 72° – 30 s), 72 °C – 3 min, 4 °C – hold. Number of cycles yielding best results was determined experimentally and was further individually adjusted for each batch of prepared cDNA. Libraries were sequenced using Illumina NovaSeq 6000.

Libraries for direct RNA sequencing (DRS) of ovarian and testicular RNA were prepared using Direct RNA sequencing kit (ONT, Cat# SQK-RNA002) following manufacturers protocol, with adjustment to magnetic beads clean-up steps: each time KAPA Pure Beads (Roche) were used and RNA-bead mixture was incubated stationary on bench at RT. To improve sequencing performance and efficiency 90–150 ng of *Saccharomyces cerevisiae* or *Saccharomyces pombe* oligo(dT)-enriched mRNA was added to all samples. Sequencing experiments were performed on MinION device and Flow Cell Type R9.4.1 (ONT, Cat# FLO-MIN106D) with MinKNOW 19.10.1 (MinKNOW core 3.5.5; Bream 4.2.11; GUI 3.5.10) used for data collection, and basecalling with Guppy 6.0.0 (ONT).

## PAT assay

PAT assay on ovarian and testicular RNA was performed using cDNA-PCR Sequencing kit (ONT, Cat# SQK-PCS111) following manufacturer's protocol with following modification allowing for amplification of transcript-specific product: in "Selecting for full-length transcripts by PCR" step, kit-provided cDNA Primer (cPRM) was replaced with 10 mM forward transcript-specific primer and 10 mM universal reverse primer targeting cDNA RT Adapter (CRTA) sequence used in previous steps of the protocol (see Supplementary Table 1 for primer sequences). Amplification was run as described in manufacturers protocol with annealing temperature (57 °C) and extension time (90 s) adjusted to

custom primers and expected product lengths. After amplification PCR products were run on 2% agarose gel and visualized using Che-miDoc Imaging System (Bio-Rad).

## Sequencing data analysis

To figure out poly(A) tail length DRS reads were mapped to the Gencode M26 reference transcript sequences[70] using Minimap 2.17[71] with options -k 14 -ax map-ont –secondary = no and processed with samtools 1.9 (samtools view -b -F 2320)[72] to removed supplementary alignments and reads mapping to reverse strand. All unmapped reads were discarded from analysis. For each read length of poly(A) tail were estimated using the Nanopolish 0.13.2[73] polya function and only reads tagged as PASS were considered in later analyses. Samples from the same condition were analyzed together since they were strongly corelated. Analyses of changes in median poly(A) tail length and mRNA abundance were performed using R. Supplementary Data 2, 6, and 7 files contain the number of counts, mean, median, and geometric mean poly(A) tail lengths. Genes represented by more than 20 reads, and with median poly(A) tail length difference WT:mutant less than −10 nt were selected as potential TENT5BC, TENT5C, and TENT5D targets. We took genes represented by at least 20 reads and with medians calculated for poly(A) tail lengths greater by 10 nt than in the control as potential TENT5B targets in the GFP knock-ins.

Differential expression analysis was performed by mapping reads obtained from Illumina and DRS sequencing to the mouse GRCm39 genome[74] using Minimap 2.17, with options -k 14 -ax splice -uf, features were assigned using Gencode M26. DRS datasets were processed using featureCounts[75] from the subread package in the long read, strand-specific mode (-L -s 1), including only features covered by at least 20% (−fracOverlapFeature 0.2). Illumina datasets were mapped to the mouse GRCm39 genome using STAR[76] on default program settings and processed by featureCounts in the short-read mode (-p −O) including only features covered by at least 20% (−fracOverlapFeature 0.2). R software DESeq2[77] were used to determine differences in mRNA expression levels. The shrinkage approach of DESeq2 was used to implement a regularized logarithm transformation for better visualization and ranking of genes. Supplementary Data 5 contains DESeq statistics for transcripts – Illumina reads dataset.

Motif enrichment analysis was performed independently on three sets of potential TENT5B/C/D targets (Supplementary Data 3, 4, 6, and 7) in comparison to their backgrounds. For TENT5C and TENT5D the background datasets contain the sequences of transcripts identified in the DRS data for WT, represented by more than 20 mapped reads per transcript. For TENT5B/C the background dataset contains transcripts present in WT DRS data assigned to 522 oocyte-specific genes identified based on RNA-seq dataset (Supplementary Data 5). Fasta sequences of 3′ UTRs, exons, and 5′ UTRs of potential TENT5 substrates and their background were obtained using bedtools getfasta tool (v. 2.29.2)[78], using bed files with coordinates downloaded from UCSC Table Browser tool (GENCODE M26 track and known gene table)[79] and GRCm39 genome sequence. Sequence motifs enriched in 3′ UTRs, exons, and 5′ UTRs of TENT5B/C/D potential substrates were identified using the STREME tool[80], run with options --dna --order 4.

CPEB motif identification was performed on the same fasta files like motif enrichment analysis. The motif scanning analysis was performed with the FIMO tool[80] on default settings using published CPEB1 and CPEB2 motif sequences[81]. We obtained groups of transcripts containing and not containing CPEB motifs in TENT5 B/C/D background datasets. We used the same datasets that were used in motif enrichment analysis. Then, we plotted poly(A) tail lengths distributions for those groups. We also plotted poly(A) tail lengths distributions for potential TENT5B/C targets containing and not containing CPEB motifs.

Comparison of the length of structural elements of transcripts identified as potential TENT5 B/C/D targets, were performed on fasta

files used in motif enrichment analysis and obtained using sektk comp tool (https://github.com/lh3/seqtk)[82]. Estimation of the statistical significance of differences between the 3′UTR, exons, and 5′UTR lengths of potential TENT5 targets and background were performed using the Mann–Whitney–Wilcoxon test.

Comparison of the GC content of structural elements of transcripts identified as potential TENT5 B/C/D targets, were performed on fasta files used in motif enrichment analysis and obtained using sektk comp tool (https://github.com/lh3/seqtk). Estimation of the statistical significance of differences between the GC contents of potential TENT5 targets and background was performed using the Mann–Whitney–Wilcoxon test.

To identify signal peptide presence in the proteins encoded by potential TENT5 targets, their sequences were downloaded from the UniProt Release 2023_02 database[83] and analyzed for the presence of signal peptides using TargetP Gene ontology analysis[84]. To dataset of potential TENT5D targets we added *Rnase2a*.

Gene ontology analysis was performed using the BioMart R library[82]. For heatmaps we chose 13 more abundant GO-terms assigned to potential Tent5 B/C/D targets.

## Data visualization

All data visualization was performed in R using "ggplot" library[85].

## Translation rate calculation

For every oocyte imagined and analyzed, YPET and mCherry fluorescence values were collected as described above. Oocytes with missing data were discarded altogether from analysis. YPET value was divided by mCherry value for each individual oocyte and time-point for normalization purposes. For each oocyte, linear regression model was calculated using "lm()" function with "YPET/mCherry intensity - timepoint" model in Rstudio software, and beta regression coefficient (describing slopes direction and degree) was used as a translation rate.

## Statistical analysis

All statistical analyses were performed using Rstudio. All results (except sequencing data) were compared using two-sided t-test for normally distributed data, Mann–Whitney–Wilcoxon test for non-normally distributed data (with the exception of follicle size comparison (Fig. 2E) where rank sum test artificially flattens difference between compared genotypes due to small number of significantly bigger follicles in WT ovaries), Fischer exact test for 2 × 2 contingency tables, and Chi-square test for bigger tables. Normality was checked for all datasets by Pearson normality test. P values lower or near assumed significance value of 0.05 were reported on figures. Full p values, number of samples/specimens analyzed, error bars are reported on figures and fully described in corresponding figure legends.

## Reporting summary

Further information on research design is available in the Nature Portfolio Reporting Summary linked to this article.

# Data availability

Supplementary Figs. 1–9 together with Supplementary Table 1 containing oligonucleotides, dsDNA, and sgRNA sequences, are available in Supplementary Information file. Results of sequencing data analyses are available as Supplementary Data 1–7. All DRS sequencing data generated in this study have been deposited in the European Nucleotide Archive (ENA) database under accession code PRJEB63526. The raw Illumina RNAseq data have been deposited in Gene Expression Omnibus (GEO) database under accession code GSE239661. Additionally, ENA sample accession numbers together with DRS run details are listed in Supplementary Data 1. Any additional information required to reanalyze the data reported in this paper is available from

the corresponding author upon request. Source data are provided with this paper.

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

## Acknowledgements

We thank the members of the Andrzej Dziembowski lab for their support, prof. Marco Conti (UCSF) for sharing YPET and mCherry mRNA reporter-carrying plasmids, Aleksandra Brouze for critical reading and proof-reading the manuscript, and all members of the Genome Engineering Unit of the International Institute of Molecular and Cell Biology for maintaining the animal colony and for animal genotyping. NGS was performed thanks to the Genomics Core Facility CeNT UW (RRID:SCR_022718), using the NovaSeq 6000 platform financed by the Polish Ministry of Science and Higher Education (decision no. 6817/IA/SP/2018 of 2018-04-10). This project has received funding from the European Union's Horizon 2020 Research and Innovation Programme under Grant Agreement No. 810425 (AD). The research leading to these results was funded by the Norwegian Financial Mechanism 2014–2021, No. UMO-2019/34/H/NZ3/00733 (AD). This project was

also funded by the National Science Centre, Poland, Grant No. 2019/33/B/NZ2/01773 (AD).

## Author contributions

M.B. and A.D. wrote the final version of the manuscript with contributions from A.C.-C., O.G., and M.K.-K. M.B. measured TENT5B and TENT5C expression in oocytes, performed phenotype analysis of *Tent5a* KO, *Tent5b/c* dKO, and *Tent5b*[GFP] mice, GDF9 and ZP3 immunohistochemistry staining, oocyte mRNA reporter injection, prepared RNA sequencing libraries for oocytes and ovaries, and performed PAT assays. M.B. and M.S. performed *Tent5a* KO oocyte maturation. O.G. performed blood analysis, body weight measurements, and *Tent5c* and *Tent5d* phenotype analysis. M.K.-K. performed co-cultures of germ cells and Sertoli cells, all flow cytometry experiments, and cell sorting. M.S. performed *Tent5b*[GFP] mating experiments and chromatin analysis experiments. B.T. and K.J. performed all immunohistochemistry experiments in testes. S.M. prepared RNA sequencing libraries for sperm and testes. J.G. and E.B. established all mouse lines. M.B. reviewed and analyzed all gathered data except sequencing results. A.C.-C. supported by P.K. analyzed all RNA sequencing results. M.B. and A.C.-C. prepared the figures.

## Competing interests

The authors declare no competing interests.
