## [Peer Review File · Nature Communications]

REVIEWER COMMENTS

Reviewer #1 (Remarks to the Author):

Cytoplasmic polyadenylation is an essential process in both male and female gametogenesis, while the enzymes involved in this process is not clear in mammals. In the current study, the authors comprehensively analyzed the roles of Tent5 family non-canonical poly(A) polymerases in mouse gametogenesis. Tent5b and Tent5c act redundantly in oogenesis, while both Tent5c and Tent5d are essential for spermatogenesis. Direct RNA sequencing revealed major targets for Tent5b and Tent5c in ovaries, as well as targets for Tent5c and Tent5d, showing enrichment for mRNA encoding ER targeted secreted protein. This piece of work will be a great contribution to the community. I have some suggestions for the further improvement of the manuscript.

1. what is the green signal in the upper right of Fig. 2A? Does that mean autofluorescence background? Please clearly indicate it in the figure and figure legend.
2. where is the 12 week old sample in Fig. 2B and Fig. S2A? 2B says 8-week and 5 week, while S2A is not properly labelled for age.
3. There are no scale bars for images in Fig. 2B and Fig. S2A, which makes it challenging to better interpret the data. In addition, it is not a good idea to show the same images in Fig. 2 and Fig. S2. I'd suggest just move all the information in Fig. S2 to Fig. 2.
4. The Tent5bgfp/gfp shows dominant negative effect in females but not in males. It will be good to discuss about the possible reason in the discussion.
5. In the DRS of ovary samples, the authors observed much shorter poly(A) tails for mRNAs encoding proteins for oogenesis. Is this because of different oocytes composition/stage between WT and KO ovaries?
6. Related to the above concern, I highly recommend the authors to validate some of the Tent5 direct targets, including Zp3 and Gdf9, using oocytes at the same stage collected from the WT and KO mice by PAT assay. This experiment will consolidate the conclusion that Tent5b and Tent5c indeed function in the oocyte through cytoplasmic polyadenylation.

7. The testes in Fig. 5C looks strange, please double check it. In addition, the genotypes and scale bars are missing.

8. Similar to the ovary part, I highly recommend the authors to validate some of the Tent5 direct targets, including Tppp2, Insl2, and Rnsset2, using male germ cells at the same stage collected from the WT and KO mice by PAT assay.

9. It is very interesting to find that mRNA encoding an ER targeting signaling peptide are preferred targets for Tent5s. How a signal in a protein translate into the regulation of mRNA? It will be good for the author to add a discussion about the potential mechanism underlying it.

10. Will it be possible to experimentally demonstrate that whether Tent5s are involved in CPEB dependent polyA tail elongation? The experiments do not need to be done for the current manuscript. A more through discussion about how to experimentally demonstrate it will be helpful in the current manuscript.

Reviewer #2 (Remarks to the Author):

This is an extensive and beautifully reported study exploring the role of the TENT5 B, C and D poly(A) polymerases in gametogenesis. Extensive datasets are provided that use multiple and novel in vivo models and the authors' conclusions appear to be mostly justified by the data presented. Roles for TENT5 enzymes in development are described but roles in the germline poorly understood.

Given the extensive mouse models developed and used in this study, clarity of the manuscript would be greatly improved by addition of diagrams and tables that summarise results from the different models. Further, the discussion is rather uninformative and the relevance of results and mechanisms, specifically relating to gametogenesis, needs to be discussed in more detail.

Points to address:

1) Lines 114 – 122. This paragraph is very difficult to follow. The authors do have this information in a diagram; however, a table would be better to refer to. In addition, a Summary Table containing all the discovered effects for each genotype would be very useful.

2) Line 116. Male fertility or infertility of Tent5d KO? Please clarify.

- 3) Figure 1A. Are each of these graphs meant to represent a knockout mouse mother? Or both mother and father? This needs to be clarified. Data from the double knockout line should be included. The authors report infertility for some genotypes, however, this should be presented comparatively as a graph.

- 4) Line 172. Please clarify from which line these 5-week-old female ovaries were dissected.

- 5) Figure 2B. No scale has been provided for ovaries. However, a scale bar is required, particularly as the 5 week old ovary appears so much smaller, and it is not clear whether the magnification is different or if this a true representation of size differences.

- 6) Figure 2C. Is the p value reporting complete here? Only 4 significant differences are reported, however, there appear to be differences between some other columns. The caption for this graph states this is for the double KO data, however, the graph contains various genotypes and this should be clarified.

- 7) Lines 246-248. This sentence appears to be contradictory (comparing TENT5B-GFP to GFP-TENT5B), please clarify. In a related point, why might TENT5B-GFP be a gain of function mutant (Lines 350-351)? This should be explored in more detail.

- 8) Line 346. It is understandable that the authors look specifically for transcripts for oocyte-specific genes with different lengths of poly (A) tails. However, were there other transcriptomic differences between the mouse lines, that may indicate additional effects/mechanisms of TENT5 enzymes? This should be explored and described in detail.

- 9) Figure 4C. The reported p-values seem incomplete. For example, is Tent5b wt/gfp significantly different than Tent5b-/- Tent5c -/- (It is unclear which dataset the significance line refers to)?

- 10) Figure 4G. It is important to include immunostaining for ZP3 to improve analysis of the knockout ovaries. In addition, the authors should provide a table of affected oocyte-associated genes with their poly(A) tail length listed.

- 11) Section starting line 462. Does the data support these conclusions? Some differences seem non-significant? For example, Fig. 5E.

12) Lines 466-467. The terms 1C and 4C are defined in the methods, but since the results text comes prior to methods, please provide these definitions here as well.

13) Line 483 to 485. Poly(A) tail length changes from 103 to 113 during ageing. Does this represent a substantial/dramatic change as claimed? Why does this occur and how does this relate to data throughout the manuscript?

14) Figure 5F. In this graph it is unclear what the yellow and blue columns represent. A legend would improve clarity.

15) Figure 5I, J and K. Why is the graph in 5K a different format than in 5I and J, when it is depicting related information?

16) Line 566. Why do the authors look in juveniles for TENT5D target RNA features and adults for TENT5C target features? This should be clarified and detailed in the text.

17) Line 583. Are the stated 13 of 55 substrates from both TENT5B and TENT5C substrates grouped together?

18) Lines 587-601. This paragraph is intended to solidify the authors' claims that the endoplasmic reticulum-targeting signal peptide regulates TENT5 substrate recognition. The justification for this and methods described are complex and difficult to follow. This section requires more explanation. How does the data demonstrate that TENT5 polymerases are mechanistically connected to the ER signal peptide? Should this section be moved to the Discussion?

19) Line 681- "this result aligns" – The examples provided do not seem to follow on from the statement about the ER. This should be clarified. What do these examples have to do with substrate recognition according to markers of ER?

20) The information presented is interesting, but the manuscript would be greatly improved if additional discussion was included concerning the mechanisms of action of TENT5 enzymes based on their data. For example, what are the reasons for the differences between the effect of knocking out TENT5C and D in the testis, compared to the double KO of TENT5B and C in the ovaries? Discussion of the recent paper studying at TENT5D deficiency in spermatogonia should be

included. Importantly, a figure illustrating the authors model of TENT5 function in the germline needs to be included in the manuscript.

21) There are some spelling and formatting errors throughout the manuscript, which should be corrected. The manuscript would benefit from additional proof-reading.

Reviewer #3 (Remarks to the Author):

In this manuscript, Dziembowski and colleagues describe very much needed research in the poly(A) tail regulation field. They identify TENT5B and TENT5C as poly(A) polymerases working redundantly during oocyte growth and -contrary to GLD2- required for female fertility in mice. They also present a detailed characterization of the effects of TENT5C and D deletion in spermatogenesis. Importantly, they identify substrates of these enzymes using direct RNA sequencing. Finally, they present the interesting observation that the ER-targeting signal peptide is a feature that distinguishes at least some of the TENT5 substrates. Altogether, this is a very interesting report, and a tour-de-force in characterization of the in vivo phenotypes of several TENT5 proteins in fertility.

Some minor comments remain:

1) It is striking that polyadenylation depends on the signal peptide, yet the authors do not discuss how this might happen. Are the authors suggesting that TENT5 proteins bind to the signal peptide during translation? The authors should include their interpretation in the discussion.

2) Why is there such a strong phenotypic effect in TENT5B^{gfp/wt} mice given that the differences in the poly(A) tail length of substrates are so small?

3) In Fig 4E, I count 78 transcripts responsive to TENT5 dysfunction(not 72, as stated in page 9, lane 348). In addition, Table S4 only contains 55 substrates. Please, include the 78 substrate in the Table, indicating their poly(A) tail lengths in wt, Tent5b/5c^{-/-} and Tent5b/5c^{gfp/gfp} ovaries. Include columns to easily filter the groups shown in Fig 4E.

4) Figure 2A and 3A contain the same images. Please, indicate this in the figure legend, or remove Fig 2A.

5) Figure 2A/3A: TENT5C-GFP is expressed as strong as TENT5B-GFP, yet expression of TENT5C-GFP does not have a detrimental effect. Why? If the two proteins are redundant and can compensate each other, this is not expected.

6) Figure 3F: What do the authors mean by 'empty follicles in the cytoplasm'?. Are they referring to empty germinal vesicles?

7) Figure 4G: What happens to GDF9 expression in TENT5B^{gfp/gfp} oocytes? Is there premature expression of this protein along follicle growth, or just increased levels at the correct stages?

8) Figure 5G: Why are there 2 peaks in the poly(A) tail length distribution of 8-week old mice? The peak at about 200 A's is not seen in 3-week old mice. Please, comment.

9) Figure 6A-B: Can the authors provide more detail on how was this analysis performed? Indicate on the figure what is ovary vs testis, what is grey (rest of transcriptome?) vs blue (TENT5 targets? if so, numbers do not coincide with targets mentioned in previous figures). In the figure legend, the authors mention that 522 oocyte-enriched mRNAs were used, but this number does not appear in the figure. Comparisons are somehow 'unfair' as very different n° of transcripts are present in the grey vs blue boxplots. The authors should revise the analysis to downsample the grey group multiple times randomly and check whether they obtain similar results. Based on the current analysis, I am not convinced by the statement in page 15 (lanes 571-572) that the most distinguishing feature of TENT5 substrates is the length of the CDS. Change in length of 5' UTR seems at least as dramatic as that of the CDS.

10) Table S5: Please, show just the 522 genes mentioned in the main text in a separate page.

11) Tables S7, S8: Please, include information of poly(A) tail length in wt and in KO samples. Show all substrates in Fig 5H and include columns to easily filter the different groups.

Brouze at al., point-by-point responses to the reviewers' comments

We thank the reviewers for their overall positive opinion about our manuscript. We have conducted several additional experiments and addressed all the issues raised, significantly improving the manuscript. Therefore, we expect you will find this revised version suitable for publication.

REVIEWER COMMENTS

Reviewer #1 (Remarks to the Author):

Cytoplasmic polyadenylation is an essential process in both male and female gametogenesis, while the enzymes involved in this process is not clear in mammals. In the current study, the authors comprehensively analysed the roles of Tent5 family non-canonical poly(A) polymerases in mouse gametogenesis. Tent5b and Tent5c act redundantly in oogenesis, while both Tent5c and Tent5d are essential for spermatogenesis. Direct RNA sequencing revealed major targets for Tent5b and Tent5c in ovaries, as well as targets for Tent5c and Tent5d, showing enrichment for mRNA encoding ER targeted secreted protein. This piece of work will be a great contribution to the community. I have some suggestions for the further improvement of the manuscript.

We appreciate this positive opinion about our manuscript.

1. what is the green signal in the upper right of Fig. 2A? Does that mean autofluorescence background? Please clearly indicate it in the figure and figure legend.

According to the reviewer's suggestion, figure labels and appropriate descriptions in the figure legend were added to the revised version of the manuscript.

2. where is the 12 week old sample in Fig. 2B and Fig. S2A? 2B says 8-week and 5 week, while S2A is not properly labelled for age.

We apologize for the confusion, only 8- and 5-week-old females were examined, this error was corrected in the text. Also, Supplementary Figure 2A was reduced in favour of modifying and extending main Figure 2B (see next point).

3. There are no scale bars for images in Fig. 2B and Fig. S2A, which makes it challenging to better interpret the data. In addition, it is not a good idea to show the same images in Fig. 2 and Fig. S2. I'd suggest just move all the information in Fig. S2 to Fig. 2.

As pointed out by the reviewer, scale information was missing in those images. We retrieved proper scale information for these files, and it was added to this figure. 5-week-old ovary was properly scaled after adding scale information as well. Additionally, some panels were moved from Supplementary Figure 2A to main Figure 2B, leaving only additional, non-essential information in Supplementary Figure 2A.

4. The Tent5bgfp/gfp shows dominant negative effect in females but not in males. It will be good to discuss about the possible reason in the discussion.

This aspect is now discussed in the revised version of the manuscript.

5. In the DRS of ovary samples, the authors observed much shorter poly(A) tails for mRNAs encoding proteins for oogenesis. Is this because of different oocytes composition/stage between WT and KO ovaries?

We agree that *Tent5b/c* dKO mutation can have a secondary effects associated with the developmental arrest. However, we use young ovaries for the analysis to minimize such an effect. Further clearly visible opposite impact of the gain-of-function knock-in *Tent5b^{gfp/gfp}* on the same transcripts is a very strong suggestion that mRNAs with shorter poly(A) tails in *Tent5b/c* KO are genuine TENT5 substrates.

6. Related to the above concern, I highly recommend the authors to validate some of the Tent5 direct targets, including Zp3 and Gdf9, using oocytes at the same stage collected from the WT and KO mice by PAT assay. This experiment will consolidate the conclusion that Tent5b and Tent5c indeed function in the oocyte through cytoplasmic polyadenylation.

To provide additional validation for poly(A) measurements, we performed PAT assay on RNA samples isolated from 30-day-old wild-type and dKO ovaries, according to the material used for DRS, for *Zp3* and *Gdf9*. The results, which visibly recapitulate poly(A) tail length distribution for those transcripts, were added to the manuscript as in Figure 4D.

7. The testes in Fig. 5C looks strange, please double check it. In addition, the genotypes and scale bars are missing.

We confirm that testes' pictures in Figure 5C are correct. According to the reviewer's comment, we provide cross-section picture of a wild-type testicle with proper scale information added to the figure.

8. Similar to the ovary part, I highly recommend the authors to validate some of the Tent5 direct targets, including Tppp2, Insl2, and Rnsset2, using male germ cells at the same stage collected from the WT and KO mice by PAT assay.

Accordingly, to comment on the ovary sequencing results, PAT assays were performed on RNA isolated from whole testes for *Insl3* and isolated germ cells for *Tppp2* of adult males. These results are now included in Supplementary Figure 6I. PAT assay for *Rnaset2* in 3-week-old males' testes and germ cells was performed as well, but despite multiple attempts did not produce visible, transcript specific PCR product. This could be due to very low amounts of transcripts in the collected material, as shown by the number of *Rnaset2* specific reads obtained in DRS (Figure 5K). We believe that PAT assay results presented provide exhausting validation of our DRS results.

9. It is very interesting to find that mRNA encoding an ER targeting signaling peptide are preferred targets for Tent5s. How a signal in a protein translate into the regulation of mRNA? It will be good for the author to add a discussion about the potential mechanism underlying it.

The role of ER targeting is now thoughtfully discussed in the revised version of the manuscript.

10. Will it be possible to experimentally demonstrate that whether Tent5s are involved in CPEB dependent polyA tail elongation? The experiments **do not need to be done for the current manuscript**. A more through discussion about how to experimentally demonstrate it will be helpful in the current manuscript.

To have a potential for CPEB dependant regulations binding motifs must be present in TENT5s substrates. Therefore we started with bioinformatic analysis.

To determine a list of transcripts with CPEB motifs in ovaries, we examined the 3'UTR sequences of transcripts identified in DRS that are linked to the 522 oocyte-specific genes identified through RNAseq data on oocytes. Approximately 6.8% of the analysed sequences contained a single CPEB motif. We then compared the distributions of poly(A) tails for transcripts with and without the CPEB motif. Interestingly, the differences between the distributions were less pronounced in the group of transcripts containing either the CPEB1 or CPEB2 motif. Among 55 potential TENT5B/C targets, the CPEB1 motif was present in 2 (*Trim 61* and *Tcl1b1*), while the CPEB2 motif was found in 7 (*Khdc1b*, *Ccno*, *Aspm*, *Birc5*, *Bpgm*, *Ooep*, *Glrx*). This blurring effect on poly(A) tail distributions between mutants was even more pronounced in the group of potential TENT5B/C targets.

Then we conducted the same analyses on sets of transcripts associated with TENT5C and TENT5D. We examined the 3'UTR sequences of all transcripts with at least 20 reads mapped in the DRS data. In both TENT5C and TENT5D sets, only 6% of the transcripts contained the CPEB motif. None of the potential TENT5D targets had the CPEB1 motif, and the CPEB2 motif was only found in *Spata6*. Among the 21 potential TENT5C targets, only *Tppp2* contained the CPEB1 motif, and CPEB2 was found in 3 (*Gca*, *Wdr38*, and *Mtmr10*).

We concluded that it is very highly unlikely that CPEB plays a significant role in TENT5 mediated poly(A) tail elongation. The analysis of CPEB binding motifs in TENT5s substrates is now included in the revised version of the manuscript (Figure 6 and Supplementary Figure 8).

Reviewer #2 (Remarks to the Author):

This is an extensive and beautifully reported study exploring the role of the TENT5 B, C and D poly(A) polymerases in gametogenesis. Extensive datasets are provided that use multiple and novel in vivo models and the authors' conclusions appear to be mostly justified by the data presented. Roles for TENT5 enzymes in development are described but roles in the germline poorly understood.

Given the extensive mouse models developed and used in this study, clarity of the manuscript would be greatly improved by addition of diagrams and tables that summarise results from the different models.

We greatly appreciate the feedback provided by the reviewer. Regarding general remarks on manuscripts clarity, table summarising phenotypes of all mutations was added in the first chapter. Additionally, a discussion is now supplemented with extensive figure representing gametogenesis stages affected by different mutations along with the proposed model of TENT5 activity in the cell.

Further, the discussion is rather uninformative and the relevance of results and mechanisms, specifically relating to gametogenesis, needs to be discussed in more detail.

The discussion section of the revised version of the manuscript has been rewritten.

Points to address:

*1) Lines 114 – 122. This paragraph is very difficult to follow. The authors do have this information in a diagram; however, **a table would be better to refer to**. In addition, a Summary Table containing all the discovered effects for each genotype would be very useful.*

Diagram in Figure 1B was replaced with a simpler table showing the fertility status of all genotypes and sex combinations. Additionally, a summary table (Table 1) was added after Figure 1, containing all observed fertility phenotypes with distinction of genotype and sex.

2) Line 116. Male fertility or infertility of Tent5d KO? Please clarify.

A mistake has been corrected in the text: was “fertility”, should be “infertility”.

3) Figure 1A. Are each of these graphs meant to represent a knockout mouse mother? Or both mother and father? This needs to be clarified. Data from the double knockout line should be included. The authors report infertility for some genotypes, however, this should be presented comparatively as a graph.

Figure 1A represents the genotype distribution of pups born from heterozygotic matings (in the case of double knockout parents' genotype was Tent5b^{-/-} Tent5c^{+/-}) to show that mice with homozygotic mutations are born with expected Mendelian frequency. This was clarified in the text, and Figure 1A was supplemented with double knockout information.

4) Line 172. Please clarify from which line these 5-week-old female ovaries were dissected.

Proper information on the genotype was added.

5) Figure 2B. No scale has been provided for ovaries. However, a scale bar is required, particularly as the 5 week old ovary appears so much smaller, and it is not clear whether the magnification is different or if this a true representation of size differences.

Proper scale information was retrieved and added to the figure, and 5-week-old ovary was properly scaled to fit other pictures. Additionally, some of the panels were moved from Supplementary Figure 2A to Figure 2B, leaving only non-essential panels as a supplement.

6) Figure 2C. Is the p-value reporting complete here? Only 4 significant differences are reported, however, there appear to be differences between some other columns. The caption for this graph states this is for the double KO data, however, the graph contains various genotypes and this should be clarified.

All significant differences (p-value < 0,05) were added to the graph to fulfil the reviewers' request. It was also clarified in the figure description that data comes from females of different *Tent5b* and *Tent5c* genotypes.

7) Lines 246-248. This sentence appears to be contradictory (comparing *TENT5B-GFP* to *GFP-TENT5B*), please clarify. In a related point, why might *TENT5B-GFP* be a gain of function mutant (Lines 350-351)? This should be explored in more detail.

This section was rewritten to make it more clear. We also provided additional explanation on the matter in the Discussion section

8) Line 346. It is understandable that the authors look specifically for transcripts for oocyte-specific genes with different lengths of poly (A) tails. However, were there other transcriptomic differences between the mouse lines, that may indicate additional effects/mechanisms of *TENT5* enzymes? This should be explored and described in detail.

We agree that transcriptomic differences between the mouse lines may indicate additional effects/mechanisms of *TENT5* enzymes, but our data do not allow us to explore it properly. DRS sequencing method is characterized by a relatively low coverage. The number of reads is significantly smaller than RNA sequencing made on Illumina platform. We performed differential expression analysis using Deseq2 on DRS datasets. We found only 4 genes significantly downregulated (*Rpl26*, *Pramel51*, *Gm8130*, *Hba-a2*; see plot below). Deregulation of these genes shouldn't have an impact on the length of poly(A) tails. Two of them are just putative genes. We are aware that regulation of *TENT5* enzymes is a complex mechanism, but a more in-depth analysis was out of the scope of this manuscript.

Differential expression analysis Tent5b $-/-$ Ten5c $-/-$ vs Tent5b $+/+$ Ten5c $+/+$

9) Figure 4C. The reported p-values seem incomplete. For example, is Tent5b wt/gfp significantly different than Tent5b $-/-$ Tent5c $-/-$ (It is unclear which dataset the significance line refers to)?

Additional p-values were added to the graph, and existing p-value bars were corrected to clearly show to which data sets they refer.

10) Figure 4G. It is important to include immunostaining for ZP3 to improve the analysis of the knockout ovaries.

We agree on the importance of ZP3 immunochemistry. After technical difficulties, we were able to produce satisfactory staining. These results are now added to Figure 4H. In addition, the entire immunohistochemistry panel was reviewed – only a single essential image per genotype and protein was left in the main figure, and additional pictures of GDF9 and ZP3 staining were moved to Supplementary Figure 4. This should provide the reader with a better first glance at the results.

In addition, the authors should provide a table of affected oocyte-associated genes with their poly(A) tail length listed.

We added worksheets containing 522 oocyte-specific genes to Supplementary Table 6, along with statistics counted by differential polyadenylation analysis. New tables contain information about the

number of reads, the mean length of poly(A) tails, the difference in mean poly(A) tail lengths between control and mutant samples, and the p-values based on which we assessed the statistical significance of the differences.

11) Section starting line 462. Does the data support these conclusions? Some differences seem non-significant? For example, Fig. 5E.

In the experiment presented in Figure 5E only one male of each genotype was used at every timepoint, due to a limited number of KO males available. For this reason, we do not provide statistical comparison between males of the same age but present our conclusion regarding changes in germ cell populations for each genotype. An additional sentence was added to that section clarifying this experiment's design, along with an introduction of definitions for different germ cell populations.

12) Lines 466-467. The terms 1C and 4C are defined in the methods, but since the results text comes prior to methods, please provide these definitions here as well.

This section was rewritten to introduce simple definitions of 1C and 4C populations.

13) Line 483 to 485. Poly(A) tail length changes from 103 to 113 during ageing. Does this represent a substantial/dramatic change as claimed? Why does this occur and how does this relate to data throughout the manuscript?

We agree that this was an overstatement. This section was rewritten to focus on the change of total poly(A) tail length distribution, not just mean length, as it does not represent complex changes in this distribution. We additionally explain in the following lines that a more thorough examination of the influence of age on activity of TENT5 proteins was out of the scope of this work.

14) Figure 5F. In this graph it is unclear what the yellow and blue columns represent. A legend would improve clarity.

Legend was added for the clarity of the graph.

15) Figure 5I, J and K. Why is the graph in 5K a different format than in 5I and J, when it is depicting related information?

Graphs 5I, J and K share data type formatting – in all graphs, individual points represent individual reads analysed, but due to very high point density in graphs 5I and 5J this can be obstructed and seem different than graph 5K. This is now pointed out in the figure legend to make the visual difference understandable. Additionally, colours on panels 5K and 5N were unified to make the distinction of genotypes clearer.

16) Line 566. Why do the authors look in juveniles for TENT5D target RNA features and adults for TENT5C target features? This should be clarified and detailed in the text.

Additional information was added in that line to explain to the reader that in the case of TENT5D KO, rapid testes tissue deterioration occurs, so RNA from testes of young TENT5D KO males was used for sequencing. This is also mentioned when first describing DRS for testes.

17) Line 583. Are the stated 13 of 55 substrates from both TENT5B and TENT5C substrates grouped together?

We admit that the cited section of the manuscript may be confusing to the reader. In this study, we do not determine separate lists of potential TENT5B and TENT5C targets. Based on the DRS data obtained for the *Tent5B*^{-/-} *Tent5C*^{-/-} and *Tent5b*^{gfp/gfp} mutants, we determined a group of 55 potential TENT5B/C targets. Following the reviewers' suggestions, we changed the DRS data annotation method and recalculated all analyses. This affected the lists of potential TENT5B/C/D targets. We identified a peptide signal in 9 of the 55 potential TENT5B/C targets. This section was corrected in the manuscript to avoid misunderstanding.

18) Lines 587-601. This paragraph is intended to solidify the authors' claims that the endoplasmic reticulum-targeting signal peptide regulates TENT5 substrate recognition. The justification for this and methods described are complex and difficult to follow. This section requires more explanation. How does the data demonstrate that TENT5 polymerases are mechanistically connected to the ER signal peptide? Should this section be moved to the Discussion?

This section was entirely rewritten to provide better explanation of experiment's design and conclusions drawn from it.

The proposed mechanism of TENT5 proteins action in relation to signal peptide is now also thoroughly discussed in Discussion section.

19) Line 681- "this result aligns" – The examples provided do not seem to follow on from the statement about the ER. This should be clarified. What do these examples have to do with substrate recognition according to markers of ER?

This aspect is now discussed in the rewritten version of the Discussion section.

20) *The information presented is interesting, but the manuscript would be greatly improved if additional discussion was included concerning the mechanisms of action of TENT5 enzymes based on their data. For example, what are the reasons for the differences between the effect of knocking out TENT5C and D in the testis, compared to the double KO of TENT5B and C in the ovaries? **Discussion of the recent paper studying at TENT5D deficiency in spermatogonia should be included.** Importantly, a figure illustrating the authors model of TENT5 function in the germline needs to be included in the manuscript.*

Accordingly, to multiple suggestions from all reviewers, the discussion underwent profound rewriting. It now includes multiple points not discussed previously and is supplemented with the new figure presenting proposed model of TENT5s function in different germ cells.

21) *There are some spelling and formatting errors throughout the manuscript, which should be corrected. The manuscript would benefit from additional proof-reading.*

The revised version of the manuscript has been proof-read

Reviewer #3 (Remarks to the Author):

In this manuscript, Dziembowski and colleagues describe very much needed research in the poly(A) tail regulation field. They identify TENT5B and TENT5C as poly(A) polymerases working redundantly during oocyte growth and -contrary to GLD2- required for female fertility in mice. They also present a detailed characterization of the effects of TENT5C and D deletion in spermatogenesis. Importantly, they identify substrates of these enzymes using direct RNA sequencing. Finally, they present the interesting observation that the ER-targeting signal peptide is a feature that distinguishes at least some of the TENT5 substrates. Altogether, this is a very interesting report, and a tour-de-force in characterization of the in vivo phenotypes of several TENT5 proteins in fertility.

We are thankful for positive assessment of our reporting. We've put all possible effort in addressing following concerns of the reviewer.

Some minor comments remain:

1) It is striking that polyadenylation depends on the signal peptide, yet the authors do not discuss how this might happen. Are the authors suggesting that TENT5 proteins bind to the signal peptide during translation? The authors should include their interpretation in the discussion.

This aspect is now discussed in the revised version of the manuscript

2) Why is there such a strong phenotypic effect in TENT5B^{gfp/wt} mice given that the differences in the poly(A) tail length of substrates are so small?

This aspect is now discussed in the revised version of the manuscript.

3) In Fig 4E, I count 78 transcripts responsive to TENT5 dysfunction(not 72, as stated in page 9, lane 348). In addition, Table S4 only contains 55 substrates. Please, include the 78 substrate in the Table, indicating their poly(A) tail lengths in wt, Tent5b/5c^{-/-} and Tent5b/5c^{gfp/gfp} ovaries. Include columns to easily filter the groups shown in Fig 4E.

In response to reviewers' suggestions, we unified the annotation of DRS data between analysis. This resulted in changes in the numbers of reads assigned to individual genes and thus changes in the distribution of poly(A) tails lengths. We repeated differential polyadenylation analyses and selected new groups of genes which products are potential targets of TENT5B/C. We distinguished 67 genes which products have elongated poly(A) tails in *Tent5b^{gfp/gfp}* or shortened tails in *Tent5b^{-/-} Tent5c^{-/-}*. From this group we removed targets for which poly(A) tails were elongated by at least 10A in *Tent5b^{-/-} Tent5c^{-/-}* or shortened in *Tent5b^{gfp/gfp}* samples. Finally, representatives of 55 genes remained in the set of transcripts regulated by TENT5B/C. This collection overlaps with the collection of potential TENT5B/C targets described in the original manuscript.

We converted Supplementary Table 4 into two tables containing poly(A) tail length statistics and information which genes were qualified as potential TENT5B/C targets. Supplementary Table 4

contains the results of the differential polyadenylation analysis performed for the WT, *Tent5b^{gfp/gfp}* pair. Supplementary Table 5 contains the results of the differential polyadenylation analysis performed for the WT, *Tent5b^{-/-} Tent5c^{-/-}* pair. We also prepared analogous tables for potential TENT5C and TENT5D targets identified from DRS data from testis (Supplementary Tables 7 and 8). We corrected the numerical values in the manuscript and figures.

4) Figure 2A and 3A contain the same images. Please, indicate this in the figure legend, or remove Fig 2A.

As the presence of both figures is important for the flow of the manuscript, duplicated images on Figure 3A were replaced with images of different oocytes to avoid confusion.

5) Figure 2A/3A: TENT5C-GFP is expressed as strong as TENT5B-GFP, yet expression of TENT5C-GFP does not have a detrimental effect. Why? If the two proteins are redundant and can compensate each other, this is not expected.

This aspect is now discussed in the revised version of the manuscript.

6) Figure 3F: What do the authors mean by 'empty follicles in the cytoplasm'? Are they referring to empty germinal vesicles?

This sentence was correct to better explain observed phenomena.

7) Figure 4G: What happens to GDF9 expression in TENT5B^{gfp/gfp} oocytes? Is there premature expression of this protein along follicle growth, or just increased levels at the correct stages?

We agree to a great validity of that question, however we were unable to perform immunohistochemical staining of GDF9 and ZP3 proteins in *Tent5b^{wv/gfp}* and *tent5b^{gfp/gfp}* ovaries due to extremely low birth rate of those females. Even heterozygotic mutation greatly diminishes female fertility, and after phenotype description all females born in this mice line were in the first instance used for transcriptomic analyses. This is now clearly stated in the text to avoid readers confusion on missing staining.

Additionally, this figure now includes staining of ZP3, and entire figure was reduced, now showing only essential results. Full set of images presenting staining in follicles of different growth stages was moved to the new Supplementary Figure 4).

8) Figure 5G: Why are there 2 peaks in the poly(A) tail length distribution of 8-week old mice? The peak at about 200 A's is not seen in 3-week old mice. Please, comment.

The deeper study of impact of age on poly(A) tail length distribution and TENT5 proteins' activity was outside the scope of our work. This information with general observation description was added in appropriate paragraph. This aspect is now also discussed in the revised version of the manuscript.

9) Figure 6A-B: Can the authors provide more detail on how was this analysis performed? Indicate on the figure what is ovary vs testis, what is grey (rest of transcriptome?) vs blue (TENT5 targets? if so, numbers do not coincide with targets mentioned in previous figures). In the figure legend, the authors mention that 522 oocyte-enriched mRNAs were used, but this number does not appear in the figure. Comparisons are somehow 'unfair' as very different n^o of transcripts are present in the grey vs blue boxplots. The authors should revise the analysis to downsample the grey group multiple times randomly and check whether they obtain similar results. Based on the current analysis, I am not convinced by the statement in page 15 (lanes 571-572) that the most distinguishing feature of TENT5 substrates is the length of the CDS. Change in length of 5' UTR seems at least as dramatic as that of the CDS.

Following this suggestions, we decided to change the way of annotating DRS data. To this end we repeated all the comparative bioinformatic analyses of TENT5s targets. TENT5D targets were excluded this time because of their insufficient number to gain reliable results.

Numbers of potential TENT5B/C/D targets were established at a gene level (depicted on Figures 4 and 5). This produced clearer results which are easier to interpret by the reader. However all analyses based on sequence had to be done on a level of individual transcripts which number is different than the genes. Thus to determine groups of potential TENT5B/C/D targets on transcript level we included all transcripts identified in the DRS data that were mapped to genes described as potential TENT5B/C/D targets. TENT5C and TENT5D targets sets were compared with all transcripts found in DRS data of sufficient WT. TENT5B/C targets set were compared with dataset based on 522 oocyte specific genes determined based on Deseq2 analysis of RNAseq data from oocytes. We took a list of 522 genes and assigned to them transcripts found in DRS dataset for WT. Therefore, based on transcripts lists we extracted from <https://genome.ucsc.edu/cgi-bin/hgTables> bad files with coordinates. Then, we used this bad file to extract fasta files from genome.

We are aware of the differences in the number of 3'UTR and 5'UTR sequences. A common feature of transcripts is variation in the number and length of UTR sequences. Including certain variants in the transcriptome is closely related to the depth and quality of the RNA sequencing data sets used. We have selected for analysis the most recently annotated mouse genome GRCm39/mm39.

Analyses of UTR and exon lengths, as well as GC content, were performed to find distinctive features of potential TENT5B/C/D targets. We noticed some statistically significant dependencies, but we agree that differences in the size of the sets weaken the reliability of these observations. We decided to move the results of this analysis from Figure 6 to Supplementary Figure 7 as a complementary study. We also modified the paragraph of the manuscript which describes distinctive features of TENT5B/C/D targets. Material and methods were updated accordingly.

10) Table S5: Please, show just the 522 genes mentioned in the main text in a separate page.

According to your suggestions, we changed Supplementary table 5. Now Supplementary Table 6 contains data obtained from differential expression analysis performed with the Deseq2 tool. We considered genes which $\log_2(\text{baseMean}) \geq 10$ to be oocyte specific. They are presented on a separate table, which is a subpage of Supplementary Table 6.

11) Tables S7, S8: Please, include information of poly(A) tail length in wt and in KO samples. Show all substrates in Fig 5H and include columns to easily filter the different groups.

In accordance with your suggestions, we have replaced Supplementary Tables 7 and 8 with tables including the results of differential deadenylation analysis (Supplementary Table 7, 8). Now they contain information about the number of reads, the mean length of poly(A) tails, the difference in mean poly(A) tail lengths between control and mutant, and the p-values based on which we assessed the statistical significance of the differences. We identified potential TENT5C or TENT5D targets as hits with statistically significantly shortened poly(A) tails, where the length difference between the control and mutant is at least 10 adenosines. Tables include information to which group (from Venn plot in Figure 5H) the gene was classified.

REVIEWERS' COMMENTS

Reviewer #1 (Remarks to the Author):

My concerns have been adequately addressed. In addition, the authors provided strong evidence that Tent5 mediated cytoplasmic polyadenylation in the oocytes is very likely CPEB independent which is interesting. I look forward to the published version of this very nice paper.

By the way, it would be helpful to check the figure labels for compliance with formatting guidelines. For instance, genotypes in Fig. 4D and Fig. 5G should be italicized; "t" missing for "transcripts" in Fig. S8B.

Reviewer #2 (Remarks to the Author):

The authors have made substantial revisions to the manuscript and addressed my key concerns. The manuscript is greatly improved and will be of interest to the field.

As a minor comment, in the new Table 1, how was fertility assessed? For example, litter sizes and number of litters per breeding pair compared to controls. These details should be clarified in the methods.

Reviewer #3 (Remarks to the Author):

The authors have addressed my comments. I have nothing to add except that there are spelling and expression mistakes in the text. The manuscript will benefit from an English corrector.

Brouze at al., point-by-point responses to the reviewers' comments

We thank the reviewers for their feedback on the final version of the manuscript. The manuscript was corrected to remove final minor errors according to reviews and editor's comments. Below, we address all final minor points raised by the reviewers.

REVIEWER COMMENTS

Reviewer #1 (Remarks to the Author):

My concerns have been adequately addressed. In addition, the authors provided strong evidence that Tent5 mediated cytoplasmic polyadenylation in the oocytes is very likely CPEB independent which is interesting. I look forward to the published version of this very nice paper.

By the way, it would be helpful to check the figure labels for compliance with formatting guidelines. For instance, genotypes in Fig. 4D and Fig. 5G should be italicized; "t" missing for "transcripts" in Fig. S8B.

All figures were corrected for formatting errors. Labels on mentioned figures were corrected.

Reviewer #2 (Remarks to the Author):

The authors have made substantial revisions to the manuscript and addressed my key concerns. The manuscript is greatly improved and will be of interest to the field.

As a minor comment, in the new Table 1, how was fertility assessed? For example, litter sizes and number of litters per breeding pair compared to controls. These details should be clarified in the methods.

A short disclaimer on how infertility was first identified was added to the appropriately named section in Methods chapter.

Reviewer #3 (Remarks to the Author):

The authors have addressed my comments. I have nothing to add except that there are spelling and expression mistakes in the text. The manuscript will benefit from an English corrector.

The manuscript was submitted for professional language editing prior to this re-submission.